# Constitutive activation of cellular immunity underlies the evolution of resistance to infection in *Drosophila*

**Alexandre B Leitão[1†]\*, Ramesh Arunkumar[1†], Jonathan P Day[1], Emma M Geldman[1], Ismaël Morin-Poulard[2], Michèle Crozatier[2], Francis M Jiggins[1]\***

[1]Department of Genetics, University of Cambridge, Cambridge, United Kingdom; [2]Centre de Biologie du Développement, Centre de Biologie Intégrative, University Paul Sabatier, Toulouse, France

**Abstract** Organisms rely on inducible and constitutive immune defences to combat infection. Constitutive immunity enables a rapid response to infection but may carry a cost for uninfected individuals, leading to the prediction that it will be favoured when infection rates are high. When we exposed populations of *Drosophila melanogaster* to intense parasitism by the parasitoid wasp *Leptopilina boulardi,* they evolved resistance by developing a more reactive cellular immune response. Using single-cell RNA sequencing, we found that immune-inducible genes had become constitutively upregulated. This was the result of resistant larvae differentiating precursors of specialized immune cells called lamellocytes that were previously only produced after infection. Therefore, populations evolved resistance by genetically hard-wiring the first steps of an induced immune response to become constitutive.

**\*For correspondence:**
ac2016@cam.ac.uk (ABL);
fmj1001@cam.ac.uk (FMJ)

[†]These authors contributed equally to this work

**Competing interests:** The authors declare that no competing interests exist.

## Introduction

A fundamental division within immune systems is between induced and constitutive immune defences. Constitutive immunity is always active, even in the absence of infection. For example, uninfected individuals produce natural antibodies (*Savage et al., 2017*), immune cells, and antimicrobial peptides (AMPs) (*Lemaitre and Hoffmann, 2007*). In contrast, induced immune responses are triggered after infection, which frequently involve amplifying existing constitutive factors. This can include immune cells proliferating and differentiating, and the massive upregulation of AMPs and cytotoxic molecules (*Lemaitre and Hoffmann, 2007*). Constitutive immunity is fast, allowing hosts to avoid becoming infected altogether or to rapidly clear infections. However, immune defences can be costly, diverting limited resources from other fitness-related traits (*Bajgar et al., 2015*) or damaging the host through immunopathology (*Graham et al., 2005*; *Sadd and Siva-Jothy, 2006*). While induced immune responses only incur these costs after infection, they are always present for constitutive immunity, even in the absence of infection. This trade-off between speed and cost is predicted to govern the allocation of host investment into constitutive and induced immunity (*Boots and Best, 2018*; *Shudo and Iwasa, 2001*). Theoretical models predict that the key parameter determining which type of defence should be favoured is the frequency that hosts are exposed to infection— more frequent exposure favours constitutive defences (*Shudo and Iwasa, 2001*). In support of this, bacteria tend to evolve phage resistance by altering surface receptors when exposure is high and rely on induced CRISPR-Cas immunity when exposure is low (*Westra et al., 2015*). However, despite there being variation in the extent of constitutive vs induced immunity within populations (*Jent et al., 2019*), to our knowledge the key prediction that high rates of parasitism lead to induced immune responses becoming constitutively expressed has not been demonstrated.

To investigate these questions, we examined how the constitutive and induced cellular immune defences of *Drosophila melanogaster* evolve under high parasite pressure. Cellular immunity in *D. melanogaster* involves blood cells called hemocytes, which play the equivalent role to leukocytes in vertebrates. In uninfected larvae, there are two morphological and functional classes of hemocytes. Plasmatocytes constitute the majority of cells and have diverse functions including phagocytosis and AMP production (*Honti et al., 2014*). Crystal cells are a less abundant specialized cell type that produces the prophenoloxidase molecules PPO1 and PPO2, which are processed into enzymes required for the melanization of parasites and wound healing (*Dudzic et al., 2015*). Infection can trigger an induced response, where these cell types proliferate, and cells called lamellocytes differentiate from circulating plasmatocytes, and prohemocytes and plasmatocytes in the lymph gland (*Cho et al., 2020*; *Honti et al., 2014*). Lamellocytes are large flat cells with a specialised role in encapsulating and killing large parasites like parasitic wasps (parasitoids). When a parasitoid lays its egg into *D. melanogaster* larvae, plasmatocytes adhere to the egg. Then, as lamellocytes differentiate they create additional cellular layers known as a 'capsule'. Finally, the capsule is melanised by the activity of phenoloxidases produced in crystal cells and lamellocytes (*Dudzic et al., 2015*). The melanin physically encases the parasitoid and toxic by-products of the melanisation reaction likely contribute to parasite killing (*Nappi et al., 2009*).

There is substantial genetic variation in susceptibility to parasitoid infection within *D. melanogaster* populations (*Kraaijeveld and Godfray, 1999*), and this is associated with an increase in the number of circulating hemocytes (*Kraaijeveld et al., 2001*; *McGonigle et al., 2017*). Here, we used a combination of experimental evolution with physiological assays and single-cell RNA sequencing to examine how a high rate of parasitism by the parasitoid wasp *L. boulardi* affects constitutive and induced immune defences.

## Results

### The evolution of resistance is associated with a more reactive immune response

To investigate the evolution of the cellular immune response, we allowed genetically diverse populations of *D. melanogaster* to evolve under intense levels of parasitism. We established an outbred population from 377 wild-caught *D. melanogaster* females, and used this to find six experimental populations that were maintained at population sizes of 200 flies. In three of these populations, larvae were infected every generation, and flies that survived by encapsulating and melanising the wasp were used to establish the next generation (*Figure 1A*, High Parasitism populations). The other three populations were maintained in the same conditions but without infection (*Figure 1A*, no parasitism populations). In line with previous studies (*Fellowes et al., 1998*; *McGonigle et al., 2017*), the high parasitism populations increased in resistance over the generations (*Figure 1B*, Selection regime x Generation: $\chi^2 = 91.41$, d.f. = 1, $p<10^{-15}$). After 33 generations, a mean of 76% of flies survived infection in the high parasitism populations, compared to 10% in the no parasitism controls.

As populations evolved resistance, they melanised parasitoids at a higher rate (*Figure 1B*). This could result from increased activity of the encapsulation immune response, or because the flies are evolving to escape the effects of immunosuppressive molecules produced by the parasite. To distinguish these possibilities, we measured the ability of each population to react to an inert object. When a small volume of paraffin oil is injected into the larval haemocoel, it remains as a sphere and can be encapsulated (*Havard et al., 2009*). Populations that evolved without parasitism show a very limited reaction to the oil droplets (*Figure 1C*). In contrast, approximately half of larvae from selected populations encapsulate the oil droplet (*Figure 1C*; Selection Regime: $\chi^2 = 73.35$, d.f = 1, $p=2\times10^{-16}$). Therefore, high levels of parasitism led to the cellular immune system evolving a more active encapsulation response.

### Constitutive upregulation of immune-induced genes

Underlying induced immune responses are transcriptional responses to infection, so we examined how the hemocyte transcriptome altered in populations that evolved under high levels of parasitism. We harvested circulating hemocytes from *Drosophila* larvae and sequenced the transcriptomes of individual cells using the 10X Genomics platform. Initially, we pooled and jointly analysed RNAseq

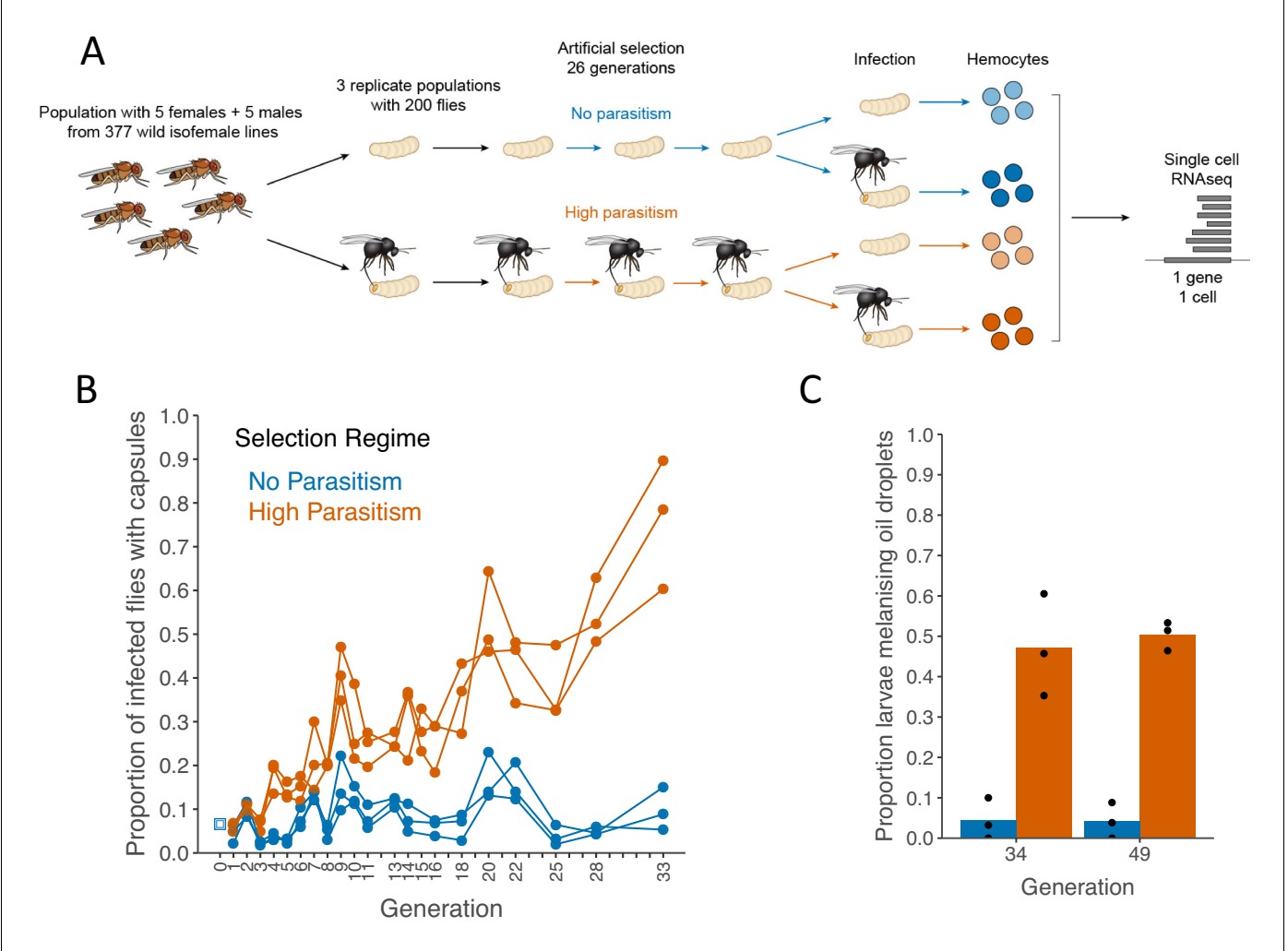

**Figure 1.** Encapsulation rate during selection for parasitoid resistance. (**A**) Schematic of the experiment (**B**) The proportion of infected larvae that gave rise to adult flies with visible capsules. An outbred population of *D. melanogaster* (source, blue square) was used to create six populations that were parasitized every generation with *L. boulardi* (high parasitism, orange) or maintained without infection (no parasitism, blue). (**C**) Proportion of larvae encapsulating oil droplets at generations 33 and 49. Dots represent the proportion calculated from 15 to 40 injected larvae in triplicate populations from each selection regime. Bar heights represent the mean per selection regime.

The online version of this article includes the following source data for figure 1:

**Source data 1.** Encapsulation rate during selection for parasitoid resistance.

reads sequenced from all cells, effectively treating them as 'bulk' RNAseq data. This allowed us to investigate global changes in hemocyte gene expression following infection and adaptation to high rates of parasitism (*Figure 2A*).

Forty-eight hours after flies were infected with parasitoid wasps there was a strong transcriptional response to infection (*Figure 2B–C*). Strikingly, when flies adapted to high parasitism conditions they showed similar transcriptional changes, even when uninfected. This can be seen by comparing the induced changes in gene expression that occur after infection to the constitutive changes in gene expression that occur after adapting to high rates of parasitism (*Figure 2B and* ). Taking the 173 genes that had at least a 50% change in transcript levels after infection or selection, in every case the direction of change was the same after infection and adaptation to high parasitism (*Figure 2B*). These genes were enriched for biological roles such as cytoskeletal organization and cell adhesion (*Supplementary file 1*). We verified that the evolved response mimicked the induced response by using quantitative PCR (qPCR) to measure the expression of the lamellocyte marker *atilla* and the anti-parasitoid effector gene *PPO3* (*Figure 2—figure supplement 1*). Again, we found

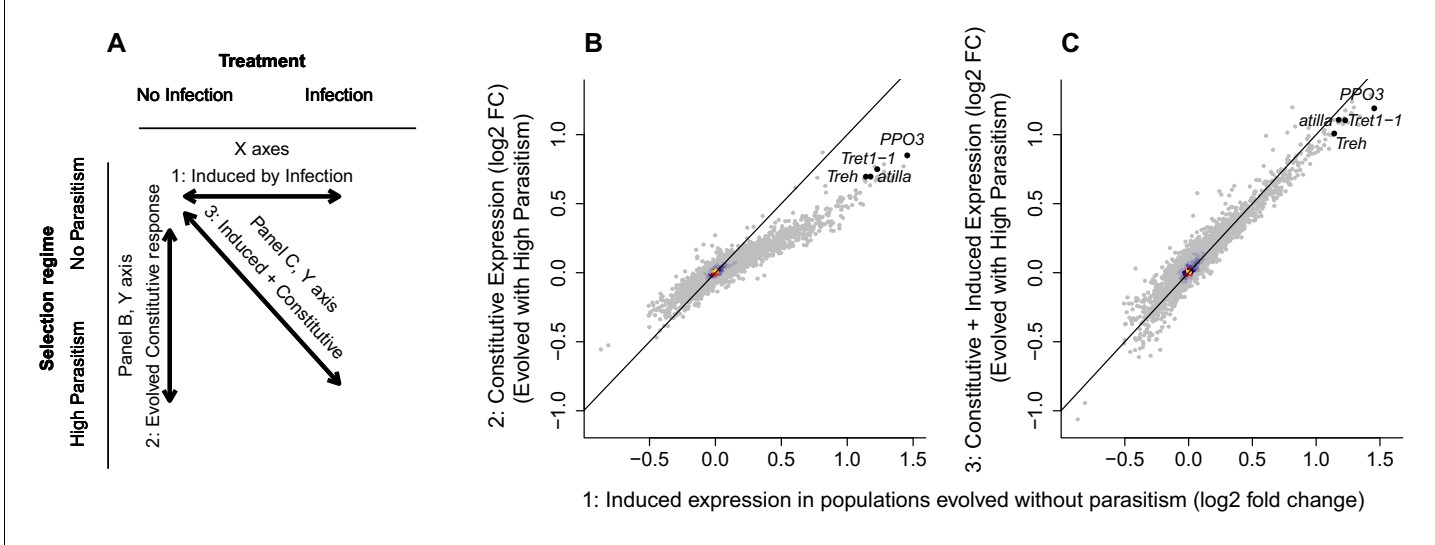

**Figure 2.** Changes in gene expression following selection for resistance and parasitoid infection. (**A**) Summary of the comparisons shown in the other panels. (**B**) The change in gene expression following infection in populations maintained without parasitoid infection (x axis) compared to the constitutive gene expression following 26 generations of selection for resistance to the parasitoid wasp *L. boulardi* (y axis). (**C**) The change in gene expression following infection in populations maintained without parasitoid infection (x axis) compared to the combined induced and constitutive change in gene expression following selection for resistance and infection (y axis). The black diagonals indicate the 1:1 line. The genes *atilla*, *PPO3*, *Trehalase* and *Tret1-1* are highlighted. Colour represents the density of overlain points. Relative expression is expressed as log$_2$(fold change). The online version of this article includes the following source data and figure supplement(s) for figure 2:

**Source data 1.** Log2 fold change in gene expression following selection and/or infection with parasitoid wasp *L.boulardi*.

**Figure supplement 1.** Gene expression in circulating hemocytes measured by qPCR.

**Figure supplement 1—source data 1.** Gene expression in circulating hemocytes measured by qPCR.

that high rates of parasitism led to these immune-induced genes becoming constitutively upregulated. Therefore, as populations evolved resistance, the induced transcriptional response to infection has become in part genetically hard-wired and constitutive.

We took the same approach to examine how the induced transcriptional response to infection changed as populations evolved resistance. We found that the transcriptional response to infection involved the same genes in the different populations (*Figure 2B* vs 2C). However, the magnitude of the changes in gene expression was substantially less in populations that had evolved under high parasite pressure (*Figure 2B* vs 2C). Therefore, the increase in constitutive gene expression has been matched by a decrease in induced expression. Again, this was verified by qPCR (*Figure 2—figure supplement 1*). This is consistent with there being a 'maximum' level of expression of these genes, so an increase in constitutive gene expression will lead to a decrease in induced expression. In support of this, combining the constitutive and induced changes in gene expression, the hemocyte transcriptome after infection is similar in the populations that evolved under high levels of parasitism and the controls (*Figure 2C*). Therefore, an induced response has become genetically fixed, as opposed to selection leading to a greater total response.

## scRNA-seq reveals novel hemocyte types

The global changes in hemocyte transcriptomes could result from a change in the relative proportion of cell types in the hemocyte population, or from a shift in gene expression within existing cell types. We therefore used the same data analysed above but investigated gene expression within individual cells. The three classical *Drosophila* hemocyte types were originally defined morphologically, but emerging single-cell transcriptomic data suggests that *Drosophila* hemocyte populations are more complex than this (*Cattenoz et al., 2020*; *Cho et al., 2020*; *Fu et al., 2020*; *Tattikota et al., 2020*). Excluding cells that did not express the pan-hemocyte markers *Hemese* or *Serpent* (*Banerjee et al., 2019*; *Bernardoni et al., 1997*), we grouped 19,344 larval hemocytes into nine clusters based on

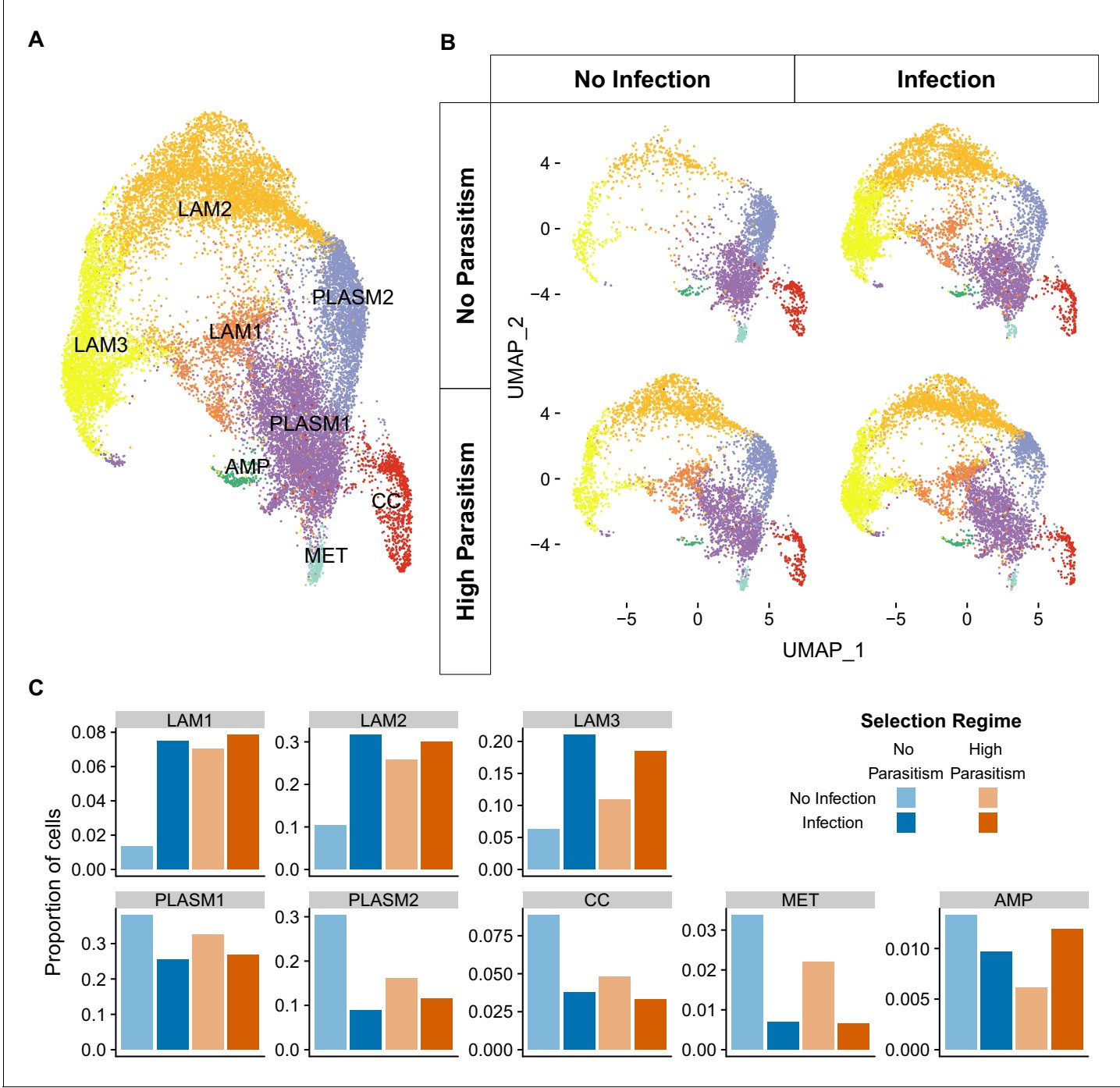

**Figure 3.** Cell states of *Drosophila melanogaster* larval hemocytes. Two-dimensional UMAP projections and cell state classification. (A) All 19,344 cells combined. PLASM: plasmatocytes; MET: metabolic plasmatocyte cluster; AMP: antimicrobial peptide plasmatocyte cluster; LAM: lamellocytes; CC: crystal cells. (B) Cells from populations that had undergone 26 generations of selection with no parasitism (top panels) or high parasitism (bottom panels), and were either uninfected (left panels) or 48 hr post infection (right panels). 64–333 million reads were sequenced in each of the eight 10X libraries that we generated (*Supplementary file 3*). The number of cells detected per library ranged from 787 to 3463 (*Figure 3—figure supplement 4*). (C) Proportion of different cell states after infection and selection.

The online version of this article includes the following figure supplement(s) for figure 3:

**Figure supplement 1.** Proportions of cells in G1, G2M, and S phase in the plasmatocyte, lamellocyte, and crystal cell clusters.

**Figure supplement 2.** Expression levels ($\log_e$) of marker genes.

**Figure supplement 3.** Comparison of scRNA-seq data sets from this paper to those from two previously published studies (*Cattenoz et al., 2020*; *Tattikota et al., 2020*).

*Figure 3 continued on next page*

**Figure supplement 4.** Number of cells and genes detected, unique molecular identifier count and percent mitochondrial gene expression per cell grouped by sample.
**Figure supplement 5.** Impact of cell number and library-specific differences on data integration, cell cluster classification and estimating cluster proportions.
**Figure supplement 6.** Flowchart of data filtering and clustering steps.
**Figure supplement 7.** Proportion of different cell states after infection and selection following data integration with 7716 anchors, the maximum number that can be detected.
**Figure supplement 8.** Expression levels (log$_e$) of marker genes from early rounds of clustering.

their transcriptomes (*Figure 3A*). Neither infection nor selection leads to the appearance of new cell states (*Figure 3B*).

We identified two large clusters of plasmatocytes. The largest was PLASM1, which we classify as the cycling plasmatocyte state as it had the greatest number of replicating cells in the G2M or S phase (*Figure 3—figure supplement 1*). The 5712 cells in this cluster had high expression of genes associated with plasmatocytes (*Col4a1*, *Pxn*, *eater*, *Hml*, *NimC1 Banerjee et al., 2019*) and actively replicating cells (*stg* [*Edgar and O'Farrell, 1990*], *polo* [*Glover, 2005*] and *scra Oegema et al., 2000*; *Figure 3—figure supplement 2A*). The next largest cluster, PLASM2, contained 2874 plasmatocytes that were enriched for the genes encoding rRNA (*Figure 3—figure supplement 2B*) or involved in the ribosome biogenesis (*Supplementary file 2*). There were two smaller plasmatocyte clusters. The AMP cluster contained 196 cells with elevated expression of genes encoding antimicrobial and antifungal peptides, including *Drs*, *Mtk* and *CecA1* (*Lemaitre and Hoffmann, 2007*; *Figure 3—figure supplement 2B*; *Supplementary file 2*). The MET cluster contained 283 cells with high expression of the heat shock proteins *Hsp24* and *Hsp27* and genes involved in organic acid metabolism (*Figure 3—figure supplement 2B*; *Supplementary file 2*). The crystal cells formed a single cluster of 906 cells (*Figure 3A*). Three clusters of cells, LAM 1, 2 and 3, had elevated expression of one or more lamellocyte marker genes (*Banerjee et al., 2019*; *Figure 3—figure supplement 2C*) and contained 1260, 5104, and 3000 cells, respectively.

Two similar scRNA-seq studies of *D. melanogaster* larval hemocytes have recently been published, so we investigated whether they identified similar clusters of cells to us (*Cattenoz et al., 2020*; *Tattikota et al., 2020*). For each of the three datasets, we assigned cells to the clusters identified in the other two studies. We found that all three analyses identified very similar clusters of crystal cells, lamellocytes and AMP-expressing plasmatocytes (*Figure 3—figure supplement 3*). However, there was less correspondence between studies for the other clusters of plasmatocytes (*Figure 3—figure supplement 3*).

## Inducible cell states are present constitutively in resistant populations

Lamellocytes are key effector cells in the anti-parasitoid immune response. In populations that evolved without exposure to parasitoids, infection resulted in a large increase in the number of cells in all three lamellocyte clusters (LAM1-3, *Figure 3B–C*). In populations that evolved under high levels of parasitism, this inducible response becomes constitutive, with uninfected larvae having 3–5 fold more cells in LAM1 and 2 (*Figure 3C*). However, this was matched by a reduction in the induced response, so the final number of cells in these clusters after infection was similar regardless of whether the populations had adapted to high or low parasitism rates (*Figure 3B–C*). These results held when replicate populations were analysed separately (*Figure 3—figure supplement 5A,B*) or when the number of cells in each library were down-sampled to match the library with the least number of cells (*Figure 3—figure supplement 5C*). Therefore, when populations evolved under high infection rates this led to the differentiation of specialised immune cells occurring in uninfected individuals.

To investigate the developmental origins of this change, we reconstructed the pathway by which plasmatocytes differentiate into mature lamellocytes. By subclustering the cells and ordering them along a pseudotemporal scale, we found a single lineage starting from the self-cycling plasmatocytes

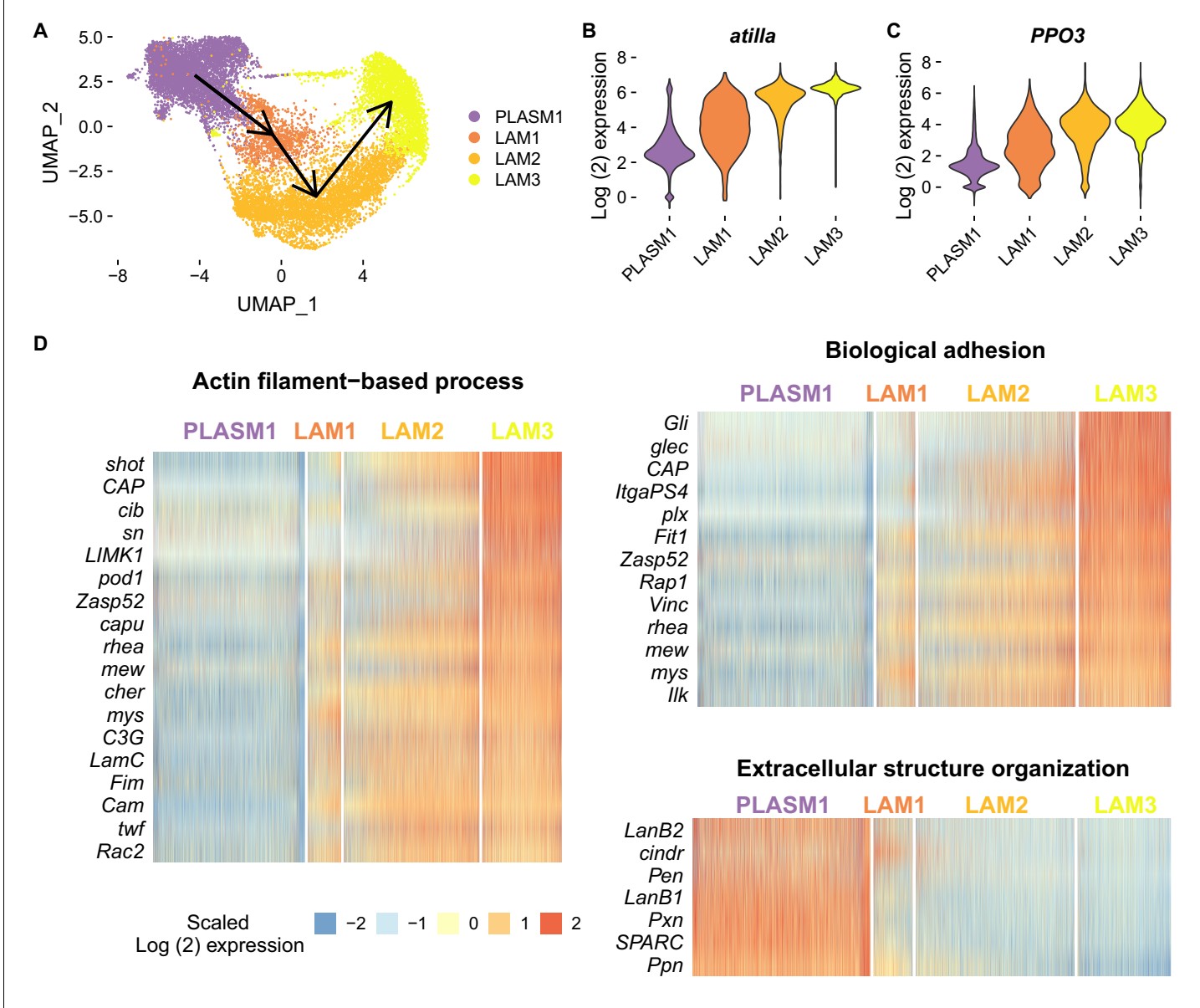

**Figure 4.** Changes in cell state following infection and adaptation to high rates of parasitism. (A) Trajectory of lamellocyte differentiation. Plasmatocyte progenitors and lamellocytes were subclustered, the trajectories inferred from multi-dimensional principle components analysis, and the lineage projected onto the two-dimensional UMAP plot. (B–C) Log$_2$ expression levels of *atilla* and *PPO3*. Relative expression levels were estimated from normalized and scaled unique molecular identifier counts. (D) Heatmaps of genes that are upregulated in either the starting or terminal clusters of lamellocyte differentiation and belonging to significantly enriched gene ontology categories (biological function). Log2 fold change gene expression levels were centred on their mean value and divided by their standard deviation.

The online version of this article includes the following source data and figure supplement(s) for figure 4:

**Source data 1.** Changes in cell state following infection and adaptation to high rates of parasitism.

**Figure supplement 1.** Trajectory of lamellocyte differentiation following subclustering.

(PLASM1) and ending in LAM3 (*Figure 4A*). Along this lineage there was a decrease in cell replication, with the largest fraction of cells in the G2M and S phase in PLASM1 and the largest fraction in G1 in LAM3 (*Figure 4—figure supplement 1A*). Importantly, controlling for the effects of cell cycle had a negligible impact on the clustering (*Supplementary file 4*) and the trajectory of lamellocyte

differentiation (*Figure 4—figure supplement 1B*). The expression of numerous genes changed as cells progressed along this trajectory (*Figure 4—figure supplement 1C*, *Supplementary file 5*), including increases in the expression of the lamellocyte effector gene *PPO3* and the lamellocyte marker *atilla* (*Figure 4B and C*). To understand the functional significance of the changes we used a gene ontology analysis. Genes involved in actin-filament based processes and biological adhesion were significantly upregulated in the terminal lamellocyte cluster, possibly reflecting the changes in cell morphology and the role of lamellocytes in capsule formation (*Figure 4D*). On the other hand, genes involved in extracellular structure organization were down-regulated as cells differentiated (*Figure 4D* and *Supplementary file 6*). This likely reflects the loss of the ability of cells to secrete extracellular matrix, which is a house-keeping function of plasmatocytes (*Bunt et al., 2010*; *Olofsson and Page, 2005*). Together, these results suggest that LAM1 and LAM2 cells are immature lamellocytes, while LAM3 cells are mature lamellocytes. Consistent with this finding, the mature lamellocyte cluster LAM3 strongly corresponds to mature lamellocyte clusters defined in previously published larval hemocyte scRNA-seq data sets (*Cattenoz et al., 2020* - LM-1; *Tattikota et al., 2020* - LM2) (*Figure 3—figure supplement 3*). Similarly, the immature lamellocyte clusters, LAM1 and 2, match the immature lamellocyte clusters in those studies (*Cattenoz et al., 2020* - LM-2; *Tattikota et al., 2020* - LM1) (*Figure 3—figure supplement 3*).

Comparing our populations, the inducible production of immature lamellocytes (LAM1 and 2) has become entirely constitutive in the populations evolved under high rates of parasitism, while the production of mature lamellocytes (LAM3) response remains largely inducible (*Figure 3C*). The differentiation of these cells explains the changes in gene expression that we observed in pooled cells after infection or selection (*Figure 1*)—168 of the 173 genes with at least a 50% change in expression in *Figure 1* also change expression in the same direction as cells differentiate from plasmatocytes into lamellocytes (PLASM1 vs LAM3). Together these results suggest that high infection rates result in the cellular immune response becoming 'primed' by constitutively producing the precursors of the cells required for parasite killing.

## Morphological differentiation of lamellocytes remains an inducible response

Lamellocytes are classically defined morphologically as large flattened cells. As expected, the proportion of morphologically identifiable lamellocytes increased after infection (*Figure 5A* and *Figure 5—figure supplement 1*; Treatment: $\chi^2$ = 48.14, d.f = 1, p=3.97×10$^{-12}$). However, in contrast to our analysis of single-cell transcriptomes, there was little difference between selection regimes in the proportion of these cells in either infected or uninfected larvae (*Figure 5A* and *Figure 5—figure supplement 1*; Selection regime x Treatment: $\chi^2$ = 0.16, d.f = 1, p=0.69; Selection regime: $\chi^2$ = 0.41, d.f = 1, p=0.52). This suggests that while the transcriptional state of cells has been constitutively activated, the change in cell morphology, corresponding to the final step of lamellocytes maturation, remains an inducible response. These cells may be equivalent to hemocytes called lamelloblasts, which morphologically resemble plasmatocytes but are induced by parasitoid infection and likely give rise to lamellocytes (*Anderl et al., 2016*). To test this hypothesis, we used microscopy to measure both cell morphology and, as a marker of the transcriptional state of the cells, atilla expression (*Figure 5—figure supplement 2*). In line with our scRNAseq data, both infection and adaptation to high parasitism rates increases the proportion of atilla positive cells (*Figure 5C*; Selection Regime x Treatment: $F$ = 25.7, d.f = 1,8, p=0.0009). However, among the atilla positive cells, infection had a large effect on cell size (*Figure 5C*; Treatment: $F$ = 41.6, d.f. = 1,7, p=0.0003), while selection had a small effect (*Figure 5C*; Selection Regime: $F$ = 4.9, d.f. = 1,7, p=0.06). Therefore, while selection has resulted in the constitutive transcriptional activation of cells, changes in lamellocyte morphology remain largely an inducible response.

## High parasitism rates lead to changes to hematopoiesis

In addition to the change in the transcriptional state of cells in circulation, we found that the populations maintained under strong parasite pressure evolved a constitutive increase in the number of hemocytes in circulation (*Figure 5B* and *Figure 5—figure supplement 1*; Selection regime: $\chi^2$ = 22.37, d.f = 1, p=2.25×10$^{-6}$). This increase in circulating hemocytes supports results from previous selection experiments (*Kraaijeveld et al., 2001*; *McGonigle et al., 2017*) and comparative

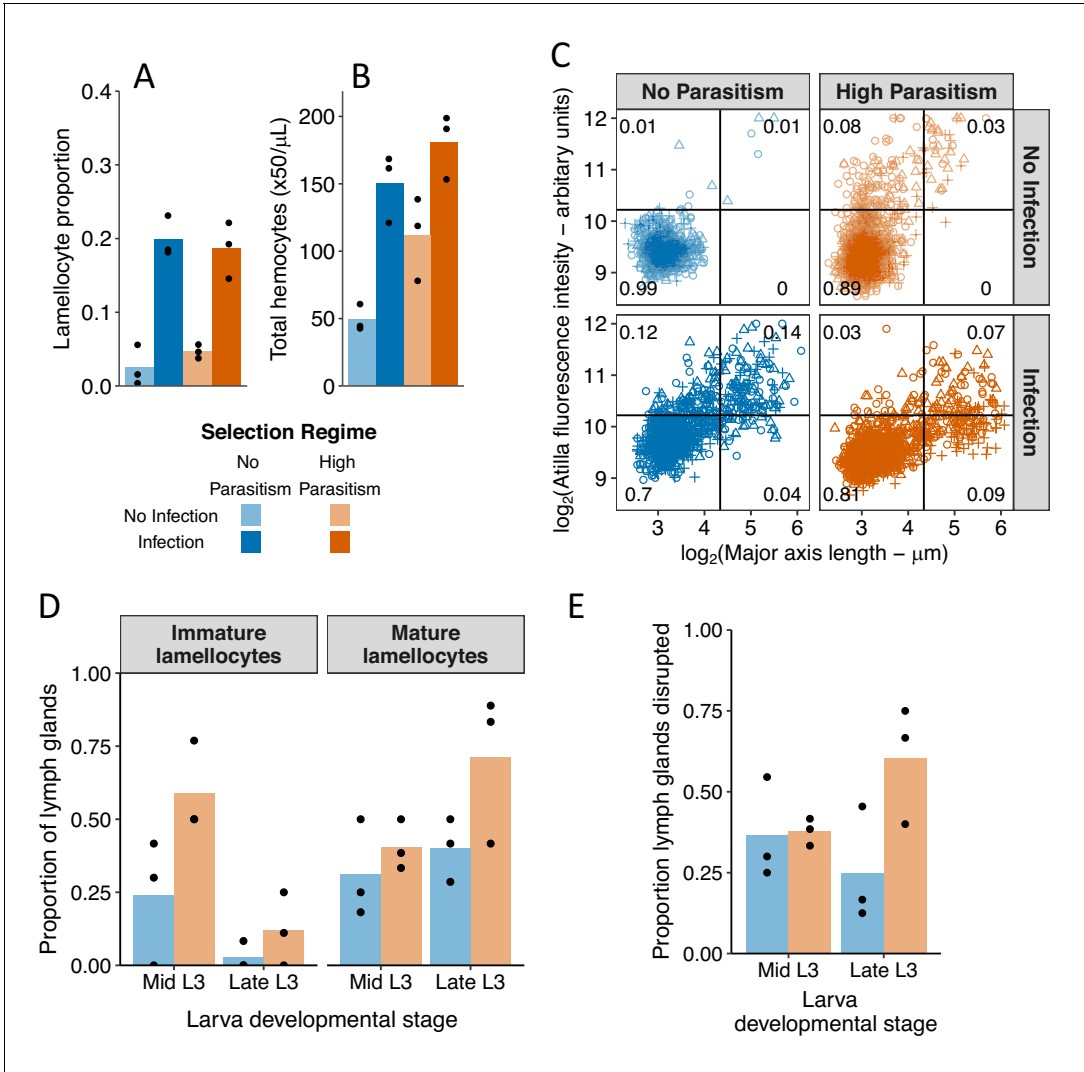

**Figure 5.** Changes in circulating and lymph gland hemocytes following infection and adaptation to high parasitism rates. (**A**) The proportion hemocytes that were morphologically identified as lamellocytes and (**B**) the concentration of total circulating hemocytes in no infection conditions (light colours) and 48 hr post infection (dark colours). Samples were collected from populations that evolved for 34 generation with high parasitism (orange bars) or no parasitism (blue bars) (**C**) Cell size and atilla expression. The expression of atilla was measured as the fluorescence intensity resulting from immunohistochemistry staining. Cell size was measured along the major axis. Replicated populations from each selection regime are represented by different symbols and the proportion of cells in each quarter of the plot is shown. (**D**) The proportion of lymph glands from uninfected larvae containing immature lamellocytes (mys$^+$ α-PS4$^-$) and mature lamellocytes (mys$^+$ α-PS4$^+$). (**E**) Proportion of disrupted lymph glands in uninfected larvae. The lymph gland anterior lobe was considered disrupted when it was either partially or completely disrupted.

The online version of this article includes the following source data and figure supplement(s) for figure 5:

**Source data 1.** Changes in circulating and lymph gland hemocytes following infection and adaptation to high parasitism rates.
**Figure supplement 1.** Changes in number of circulating hemocytes during artificial selection.
**Figure supplement 2.** Hemocyte size and atilla staining intensity in circulating hemocytes.
**Figure supplement 3.** Lamellocytes differentiation state and morphology of anterior lymph gland lobes.
**Figure supplement 4.** Hemocyte concentrations during development of 3rd instar larvae.
**Figure supplement 4—source data 1.** Hemocyte concentrations during development of 3rd instar larvae.
**Figure supplement 5.** ç Concentration of hemocytes from dissected larvae without manipulation (Circulating) and after vortexing to displace sessile hemocytes into circulation (Circulating + Sessile).
**Figure supplement 5—source data 1.** Hemocyte concentrations (circulating and circulating + sessile).
**Figure supplement 6.** Sessile crystal cells numbers.
**Figure supplement 6—source data 1.** Sessile crystal cell numbers.

analyses (*Prévost and Eslin, 1998*). *Drosophila* populations can increase the number of circulating hemocytes by relocating sessile hemocytes into circulation or by increasing the total number of hemocytes (*McGonigle et al., 2017*). We found that populations maintained under the high parasitism regime increased the total number of hemocytes, rather than move cells from the sessile compartment into circulation (*Figure 5—figure supplement 5*; Selection regime: $\chi^2$ = 34.79, d.f = 1, p=3.67×10$^{-9}$; Selection regime * Compartment: $\chi^2$ = 2.41, d.f = 1, p=0.12). In accordance with this result, the number of crystal cells in the sessile compartment was not altered in populations maintained with high parasitism (*Figure 5—figure supplement 6*; Selection regime: $\chi^2$ = 0.014, d.f = 1, p=0.91). The increase in hemocyte number is another aspect of the induced response that has become constitutive.

Lamellocytes can differentiate from sessile or circulating plasmatocytes, or from cells in the lymph gland, which is the larval hematopoietic organ. To investigate how adaptation to high parasitism conditions affected this organ, we performed immunostainings with antibodies recognizing two integrin subunits, *myospheroid* (*mys*) and the *Integrin alphaPS4 subunit* (α-PS4) (*Figure 5—figure supplement 3*). Immature lamellocytes are $mys^+$ $a\text{-}PS4^-$ whereas mature lamellocytes are $mys^+$ $a\text{-}PS4^+$ (*Louradour et al., 2017*). Mature and immature lamellocytes are detected in larvae raised with and without parasitism both in mid and late L3 stage (*Figure 5D* and *Figure 5—figure supplement 3*). However, populations that adapted to high rates of parasitism have a higher number of lymph glands containing immature lamellocytes (*Figure 5D*; main effect selection regime: $\chi^2$ = 6.90, d.f. = 1, p=0.009), and a non-significant trend towards a higher number of mature lamellocytes (*Figure 5D*; main effect selection regime: $\chi^2$ = 3.13, d.f. = 1, p=0.08). We also analysed lymph gland disruption. In the absence of parasitism, lymph glands remain intact until metamorphosis (*Grigorian et al., 2011*). In response to parasitism by wasps, the larval organ is prematurely disrupted, leading to the release of lymph gland hemocytes in circulation (*Louradour et al., 2017*). Across all our populations, in the absence of wasp parasitism, we found that at least 25% of lymph glands were disrupted (*Figure 5E* and *Figure 5—figure supplement 3*). Furthermore, in populations raised under high parasitism the proportion of disrupted lymph glands increased up to 60% at the late L3 larval stage (*Figure 5E*; selection regime*larval stage: $\chi^2$ = 3.46, d.f. = 1, p=0.06). Therefore, adaptation to high parasitism leads to an increase in both the differentiation of lamellocyte precursors in the lymph gland and the premature disruption of lymph glands. Together our analyses of both circulating hemocytes and lymph glands indicate that the *Drosophila* populations that evolved under the high parasitism regime are primed for lamellocyte differentiation.

Based on our analyses of both circulating hemocytes and lymph glands, we hypothesised that accelerated lamellocyte production during infection underlies the evolution of resistance. To investigate this hypothesis, we counted the number of hemocytes during the course of infection (*Figure 5—figure supplement 4*). In line with our earlier results, infected larvae from populations selected under high parasitism produced more hemocytes overall and more lamellocytes specifically (total hemocytes, selection regime: $\chi^2$ = 70, d.f. = 1, p<2.2×10$^{-16}$; lamellocytes, selection regime: $\chi^2$ = 14.92, d.f. = 1, p=0.11×10$^{-3}$). Supporting our hypothesis, the dynamics of lamellocyte production differs between selection regimes, with populations selected under high parasitism producing higher proportions of lamellocytes at earlier time points (*Figure 5—figure supplement 4*; Lamellocyte proportion, selection regime*time point: $\chi^2$ = 36.56, d.f. = 3, p=5.69×10$^{-8}$). At the early timepoint of 12 hr post infection, lamellocyte concentration was eight times higher in the populations that had evolved under high parasitism rates while total hemocytes was 2.5 times higher (*Figure 5—figure supplement 4*). This fast increase in circulating hemocytes numbers is likely explained by a combination of cell proliferation and relocation of sessile hemocytes into circulation, since lymph glands take longer to respond to parasitism (*Sorrentino et al., 2002*). Altogether, these results support the hypothesis that constitutive activation of immune defences allows hosts to mount more rapid immune responses.

## Discussion

Our results support the hypothesis that natural selection favours constitutive defences when infection is common (*Boots and Best, 2018*; *Shudo and Iwasa, 2001*; *Westra et al., 2015*) by demonstrating that an inducible immune response becomes constitutive when populations frequently encounter a parasite. The induced immune response to parasitoids relies on the differentiation of

lamellocytes from embryonic-derived plasmatocytes, lymph gland prohemocytes (*Honti et al., 2014*) and lymph gland plasmatocytes (*Cho et al., 2020*). Evidence for immature lamellocytes during infection has previously come from patterns of marker gene expression (*Anderl et al., 2016*; *Honti et al., 2010*) and recent scRNAseq data (*Cattenoz et al., 2020*; *Tattikota et al., 2020*). Our results show that *Drosophila* populations evolve to produce these cells constitutively if they are frequently exposed to infection. Our results contrast with populations evolved with high levels of parasitism by the parasitoid wasp *Asobara tabida*, where there was no significant overlap of differentially expressed genes between selection and infection (*Wertheim et al., 2011*). These results were generated by sequencing RNA extracted from the whole larval body (*Wertheim et al., 2011*), so it is likely they may have missed changes in gene expression that occur in only a subset of immune cells. The difference between the studies could also result from the different genetic basis of resistance to the two wasp species (*Poirie et al., 2000*).

The change in the transcriptional state of constitutive cells was not accompanied by morphological differentiation, suggesting that populations that adapt to high parasitism rates have their cellular immune defences in a state of readiness, with the final step of massive lamellocyte differentiation remaining an inducible response. Genes involved in cytoskeleton dynamics such as *βTub60D* and *shot* are among the genes with the greatest increases in expression as lamellocytes differentiate (*Figure 4—figure supplement 1C*, *Supplementary file 5*) and these may be required for the final step of changing cell shape (*Rizki and Rizki, 1994*). Furthermore, the late stages of the immune response were not qualitatively altered after selection. Populations from both selection regimes had very similar hemocyte compositions in the late stages of the response, which is similar to patterns reported from populations artificially selected with *Asobara tabida* (*Salazar-Jaramillo et al., 2017*).

Evolving resistance to parasitoid wasps is costly in *Drosophila* (*Fellowes et al., 1998*; *Kraaijeveld and Godfray, 1997*; *McGonigle et al., 2017*), so inducible immunity may be an adaptation that allows uninfected individuals to avoid this cost under low parasitism conditions. The physiological mechanism that causes the cost is unknown. It is possible that the differentiation of immature lamellocytes diverts resources from other fitness-related traits as lamellocyte differentiation is energy-demanding (*Bajgar et al., 2015*). Trehalose is the most abundant circulating sugar in insects, and it can be cleaved into glucose by the enzyme trehalase (*Mattila and Hietakangas, 2017*). In support of previous results, we found that *Trehalase* and the trehalose transporter *Tret1-1* are upregulated in hemocytes after parasitoid infection (*Figure 2*; *Bajgar et al., 2015*). Together with increased expression of glycolysis genes after infection, this is thought to redirect energy from development to immunity (*Bajgar et al., 2015*). We found that these genes are also upregulated in uninfected larvae after adaptation to high parasitism pressure (*Figure 2*). This suggests that the metabolic cost of immunity may be borne by uninfected flies in our resistant populations, potentially contributing to the cost of evolving resistance.

After infection, whether the host or parasitoid survives is a race between parasitoid development and the formation of a cellular capsule by the immune system (*Kim-Jo et al., 2019*). To increase their success, parasitoid wasps inject venom proteins and virus like particles (VLPs) during oviposition to suppress the host immune system (*Labrosse et al., 2003*). These factors affect different components of the immune system (*Poirié et al., 2014*), with the *L. boulardi* strain used in this study affecting lamellocyte differentiation (*Leitão et al., 2019*). Thus, resistant populations must circumvent the suppressive effects of the wasp venoms and VLPs, and constitutive activation of cellular immunity may be a way to achieve this by killing the parasite before immunity is suppressed.

We find that populations adapted to high levels of parasitism evolved two phenotypes that likely explain their higher resistance. Firstly, they produce more hemocytes in homeostasis, which leads to a faster increase in circulating hemocytes in early time points of infection. Secondly, they produce lamellocytes more rapidly upon infection, likely largely due to the constitutive production of immature lamellocytes. Furthermore, cytotoxic molecules could be more readily available as there is an increase in the constitutive expression of *PPO3* in populations adapted to high rates of parasitism. *PPO3* encodes an intracellular prophenoloxidase produced by lamellocytes that, when activated, is responsible for the capsule melanization (*Dudzic et al., 2015*). We conclude that the constitutive activation of the cellular immune response likely explains the greatly increased resistance of populations adapted to high rates of parasitism.

# Materials and methods

## Key resources table

| Reagent type (species) or resource | Designation | Source or reference | Identifiers | Additional information |
|---|---|---|---|---|
| Software, algorithm | GitHub | https://github.com/ | RRID:SCR_002630 | |
| Software, algorithm | LSD | *Schwalb et al., 2020* | | https://cran.r-project.org/web/packages/LSD/index.html |
| Software, algorithm | MAST | *Finak et al., 2015*; https://doi.org/10.1186/s13059-015-0844-5 | | |
| Software, algorithm | tidymodels | *Kuhn and Wickham, 2020*; https://www.tidymodels.org/. | | https://www.tidymodels.org/ |
| Software, algorithm | ranger | doi:10.18637/jss.v077.i01 | RRID:SCR_017344 | https://cran.r-project.org/web/packages/ranger/index.html Version 0.12.1 |
| Software, algorithm | R | http://www.r-project.org/ | RRID:SCR_001905 | Versions 3.6 and 4 |
| Software, algorithm | EBI database | doi: 10.1093/nar/gkz268. | RRID:SCR_004727 | |
| Software, algorithm | Sequence Read Archive | http://www.ncbi.nlm.nih.gov/sra | RRID:SCR_004891 | |
| Software, algorithm | pheatmap | https://www.rdocumentation.org/packages/pheatmap/versions/0.2/topics/pheatmap | RRID:SCR_016418 | |
| Software, algorithm | ArrayExpress | http://www.ebi.ac.uk/arrayexpress/ | RRID:SCR_002964 | |
| Software, algorithm | Gene Expression Omnibus | https://www.ncbi.nlm.nih.gov/geo/ | RRID:SCR_005012 | |
| Software, algorithm | REViGO | doi: 10.1371/journal.pone.0021800. | RRID:SCR_005825 | |
| Software, algorithm | gridExtra | *Auguie, 2017*; https://CRAN.R-project.org/package=gridExtra | | https://cran.r-project.org/web/packages/gridExtra/index.html |
| Software, algorithm | cowplot | cran | RRID:SCR_018081 | https://cran.r-project.org/web/packages/cowplot/index.html version 1.0.0 |
| Software, algorithm | tidyverse | cran | RRID:SCR_019186 | https://CRAN.R-project.org/package=tidyverse version 1.3.0 |
| Software, algorithm | limma | doi: 10.1093/nar/gkv007. | RRID:SCR_010943 | Version 3.44.3 |
| Software, algorithm | Slingshot | doi: 10.1186/s12864-018-4772-0. | RRID:SCR_017012 | Version 1.6.1 |
| Software, algorithm | KEGG | doi: 10.1093/nar/gkj102. | RRID:SCR_012773 | |
| Software, algorithm | Reactome | doi: 10.1093/nar/gkv1351. | RRID:SCR_003485 | |

*Continued on next page*

*Continued*

| Reagent type (species) or resource | Designation | Source or reference | Identifiers | Additional information |
|---|---|---|---|---|
| Software, algorithm | Cell Ranger | https://support.10xgenomics.com/single-cell-gene-expression/software/overview/welcome | RRID:SCR_017344 | Version 2.1.1 |
| Software, algorithm | Seurat | doi:10.1016/j.cell.2019.05.031 | RRID:SCR_007322 | Version 3.1.1 |
| Software, algorithm | Flymine | doi:10.1186/gb-2007-8-7-r129 | RRID:SCR_002694 | v51 2020 December |
| Strain, strain background (*Drosophila melanogaster*) | CAMOP3 | This paper | CAMOP3 | Outbred population of *D. melanogaster* |
| Antibody | anti-Atilla (mouse monoclonal) | doi: 10.1556/ABiol.58.2007.Suppl.8 | L1 | 1 in 100 |
| Antibody | anti-alphaPS4 (rabbit polyclonal) | doi: 10.1038/nature05650 | | 1 in 200 |
| Antibody | anti-Mys (mouse monoclonal) | Developmental Studies Hybridoma Bank | RRID:AB_528310C; Cat#:F.6G11c | 1 in 100 |
| Sequence-based reagent | PPO3_qPCR_2_Fw | This paper | qPCR primers | GATGTGGACCGGCCTAACAA |
| Sequence-based reagent | PPO3_qPCR_2_Rev | This paper | qPCR primers | GATGCCCTTAGCGTCATCCA |
| Sequence-based reagent | atilla_qPCR2_Fw | This paper | qPCR primers | ACCCACCAAATATGCTGAAACA |
| Sequence-based reagent | atilla_qPCR2_Rv | This paper | qPCR primers | TTGATGGCCGATGCACTGT |
| Sequence-based reagent | RpL32_qPCR_F-d | https://doi.org/10.1371/journal.ppat.1004728 | qPCR primers | TGCTAAGCTGTCGCACAAATGG |
| Sequence-based reagent | RpL_qPCR_R-h | https://doi.org/10.1371/journal.ppat.1004728 | qPCR primers | TGCGCTTGTTCGATCCGTAAC |
| Other | Phalloidin staining | Invitrogen | RRID:AB_2315633; Cat. #: A12381 | 1 in 2000 |
| Other | Mineral oil | Sigma-Aldrich | Cat#: M5904 | |

## Artificial selection

An outcrossed *D. melanogaster* population was founded from the progeny of 377 females collected in July 2018 from banana and yeast traps set up in an allotment plot in Cambridge, UK (52° 12'12.5"N 0°09'00.6"E). Single females were sorted into vials with cornmeal food (per 1200 ml water: 13 g agar, 105 g dextrose, 105 g maize, 23 g yeast, 35 ml Nipagin 10% w/v) to create isofemale lines. From the progeny of each isofemale line, 5 females and five males lines were sorted into a population cage to create the source population with 3770 flies. Flies from the source population were allowed to lay eggs overnight in 90 mm agar plates (per 1500 ml water: 45 g agar, 50 g dextrose, 500 ml apple juice, 30 ml Nipagin 10% w/v) spread with yeast paste (*Saccharomyces cerevisiae* – Sigma-Aldrich #YSC2). Eggs were removed from the agar plate with phosphate- buffered saline (PBS) and a paintbrush, collected in 15 ml centrifuge tubes and allowed to settle on the bottom of the tube. 500 µl of egg solution were transferred into a 1.5 ml microcentrifuge tube. 5 µl of egg solution were transferred into plastic vials with cornmeal food and incubated at 25°C, in a 14-hr light/10-hr dark cycle and 70% humidity. 48 hr after egg transfer, a single female wasp was added into vials for infection treatments and allowed to infect larvae for 24 hr. Vials from infection and no

infection treatments were incubated for 12 days in total. Flies from infection treatment with visible capsules were collected and randomly sorted into triplicate selection lines (NSRef1-3). Flies from no infection treatment were sorted into triplicate control populations (C1-3). Consecutive generations of selection were maintained with the same protocol described. Population sizes were maintained at 200 adult flies for all populations, at roughly 50% sex ratio.

## Estimation of encapsulation ratio

To determine encapsulation ability, 10 vials per line were prepared without infection and 20 vials with infection according to the selection protocol described above. Flies emerging from the no infection vials (control flies) were counted. Flies from infected vials were discriminated for the presence/absence of capsules by squishing anaesthetized flies between two glass slides and observing under a dissecting microscope. Encapsulation ratio was calculated according to the formula:

$$Encapsulation\ Ratio = \frac{Capsules}{Control - Uninfected}$$

In the formula, Capsules represent the mean number of flies from infection vials with visible capsules and Uninfected represent the mean number of flies with no discernable capsule. Control represent the mean number of flies from vials with no infection. With this method we have an estimate of proportion of infected flies with successful encapsulation per line.

## Wasp maintenance

Leptopilina boulardi strain NSRef (*Varaldi et al., 2006*) was maintained in a very susceptible *D. melanogaster* outbreed population named CAMOP2. Cornmeal vials were prepared with 6 µl of eggs as described above. Two female wasps and one male were added to each vial. Vials were incubated for 24 days at 25˚C, in a 14-hr light/10-hr dark cycle and 70% humidity. Adult wasps were collected and maintained in cornmeal vials with a drop of honey.

## Oil injection

Egg solutions were prepared in 1.5 ml microcentrifuge tubes as described above. 15 µl of eggs were transferred into 50 mm diameter plastic plates with cornmeal food. Plates were incubated for 72 hr at 25˚C. Borosilicate glass 3.5' capillaries (Drummond Scientific Co. 3-000-203-G/X) were pulled to form thin needles in a needle puller (Narishige PC-10). The needle was backfilled with paraffin oil (Sigma-Aldrich #M5904) with a syringe and attached to a nanoinjector (Drummond Scientific Co. Nonoject II). Late 2nd instar and early 3rd instar larvae were carefully removed with forceps from cornmeal food plates and placed on filter paper, in groups of 20. Larvae were carefully injected with 4.6 nl of oil. After injection, $_{dd}H_2O$ was added with a brush to remove the larvae and a total of 40 larvae were transferred into a cornmeal food vial and incubated at 25˚C. After 48 hr larvae were removed with a 15% w/v sugar solution and transferred into $_{dd}H_2O$ droplets on top of a plastic plate. Larvae were observed under a dissecting microscope and scored for the presence of melanization in the oil droplet.

## Hemocyte counts

To estimate hemocyte concentration, eggs were transferred into cornmeal vials following the artificial selection protocol described above and incubated at 25˚C. In infection treatment vials, three female wasps were allowed to infect larvae for 3-hr, 48-hr post-egg transfer. This protocol guarantees a high infection ratio (*Leitão et al., 2019*) and reduces the time window for the treatment. To count hemocytes during the course of experimental evolution, vials were incubated 48 hr post infection. To estimate hemocyte numbers during the course of infection, vials were incubated for 12-, 24-, 36-, and 48-hr post-infection. Third instar larvae were collected from the food vials, washed in $_{dd}H_2O$, dried in filter paper and transferred to a multispot porcelain plate in groups of 10. Hemolymph was collected by tearing the larval cuticle from the ventral side. 2 µl of hemolymph were rapidly collected and diluted in 8 µl of Neutral Red (1.65 g/L PBS – Sigma-Aldrich #N2889) to help visualize the cells. The diluted hemolymph was transferred into a Thoma chamber and the number of hemocytes in 0.1 µl was counted. Plasmatocytes and lamellocytes were distinguished by morphology (*Rizki, 1957*). To relocate hemocytes from sessile compartments into circulation, groups of 10 larvae

were transferred into a 2 ml microcentrifuge tube with 0.5 ml water and 0.5 g of glass beads (425–600 µm, Sigma #G9268). The tubes were then vortexed at maximum speed for 1 min (*Petraki et al., 2015*) and larvae prepared for hemolymph collection as described above.

## Crystal cell counts

Third instar larvae were collected as described above, transferred to a microcentrifuge tube with 1 ml PBS and heated at 68°C for 15 min. This treatment induces the spontaneous activation of PPO in hemocytes, allowing the identification of sessile crystal cells (*Dudzic et al., 2015*). The posterior dorsal part of individual larvae was imaged and crystal cells on the A7 segment manually counted.

## Quantitative PCR

To analyse the expression of *PPO3* and *atilla*, RNA was extracted from hemocytes pooled from 50 to 70 larvae, 24-hr post infection. Larvae were cleaned in $_{dd}H_2O$, dried in filter paper and transferred into a well of a multispot porcelain plate with 100 µl of PBS. Hemolymph was collected by tearing the ventral side of the larval cuticle. Samples were homogenized in 400 µl TRIzol [Ambion #15596018] and kept at −80°C. For RNA extraction, samples were defrosted and centrifuged for 10 min at 4°C at 12.000 g. 160 µl of supernatant was transferred into 1.5 ml microcentrifuge tubes, 62.5 µl of chloroform [Fisher Scientific #C/4920/08] was added, tubes were shaken for 15 s and incubated for 3 min. After a 10 min of centrifugation at 12,000 g at 4°C, 66 µl of the aqueous phase was transferred into a 1.5 µl microcentrifuge tube, 156 µl of isopropanol [Honeywell #33539] added and the solution thoroughly mixed. After 10 min incubation samples were centrifuged for 10 min at 12.000 g at 4°C and the supernatant was removed. RNA was washed with 250 µl 70% ethanol, centrifuged for 2 min at 12.000 g at 4°C. Ethanol was removed, samples dried, 20 µl of nuclease-free water [Ambion #AM9930] was added and samples incubated at 45°C for 10 min. cDNA was prepared from RNA samples with GoScript reverse transcriptase (Promega) according to manufacturer's instructions. cDNA was diluted 1:10. Exonic primers for *D. melanogaster PPO3* and *atilla* were designed in NCBI Primer-BLAST online tool: (PPO3_qPCR_2_Fw: 5'-GATGTGGACCGGCCTAACAA-3'; PPO3_qPCR_2_Rev 5'-GATGCCCTTAGCGTCATCCA-3'; atilla_qPCR2_Fw: 5'- ACCCACCAAATA TGCTGAAACA-3'; atilla_qPCR2_Rv: TTGATGGCCGATGCACTGT). The gene *RpL32* was used to normalize gene expression (RpL32_qPCR_F-d: 5'-TGCTAAGCTGTCGCACAAATGG-3'; RpL_qPCR_R-h 5'- TGCGCTTGTTCGATCCGTAAC-3'); (*Longdon et al., 2015*). Sensifast Hi-Rox SyBr kit (Bioline) was used to perform the RT-qPCR on a StepOnePlus system (ThermoFisher Scientific). Each sample was duplicated (qPCR technical replica). The PCR cycle was 95°C for 2 min followed by 40 cycles of 95°C for 5 s, 60°C for 30 s.

## Circulating hemocytes immunohistochemistry, image acquisition, and analysis

Larvae were prepared in the same conditions as for hemocyte counts, described above. Larvae were washed in $_{dd}H_2O$, dried on filter paper and transferred into a well of multispot porcelain plate with 25 µl PBS. Hemolymph was collected by tearing the ventral cuticle with forceps and the hemocyte suspension was transferred into a reaction well of a microscope slide (Marienfeld, #1216330). Samples were incubated for 30 min in a humid chamber, fixed for 20 min with 3.8% formaldehyde (Sigma-Aldrich #F8775) and washed three times with PBS. Cells were blocked and permeabilized with 0.01% Triton-X and 1% NGS diluted in PBS for 30 min and washed three times with PBS. Primary antibody against *atilla* (L1, 1:100, *Kurucz et al., 2007*) was added with 1% NGS overnight at 4°C. Primary antibody was washed with PBS for 5 min three times and secondary antibody (Alexa Fluor 488, anti-mouse, 1:1000) was applied overnight at 4°C. Cells were washed three times with PBS and stained with Hoechst 33342 (Invitrogen #H3570, 1 µg/ml) and phalloidin (Alexa Fluor 594, Invitrogen #A12381, dilution 1:2000) for 30 min. After cells were washed three times with PBS, cover slides were mounted with Vectashield (Vector laboratories).

Images were acquired in a Leica DM6000 fluorescence microscope with a 40x objective. Images were analysed with Fiji version 1.0 (*Schindelin et al., 2012*). Cells were identified based on actin and nuclear staining and manually outlined with the selection tool. Maximum intensity of *atilla* staining and length of the major axis was recorded for each cell. The threshold for staining intensity was obtained by measuring intensity of cells that were stained following the immunohistochemistry

protocol (described above) but with no primary antibody. The threshold for cell size was obtained from an independent set of images where cells were measured and classified as lamellocytes.

## Lymph gland immunohistochemistry, image acquisition, and analysis

Lymph glands were dissected and processed as previously described (*Krzemień et al., 2007*). Antibodies used were rabbit anti-αPS4 (1/200) (*Krzemień et al., 2007*) and mouse anti-β Integrin (Mys) (Developmental Studies Hybridoma Bank, CF.6G11c, 1/100) (*Irving et al., 2005*). Immunostainings were performed as previously described (*Louradour et al., 2017*). Nuclei were labelled with DAPI. Optimized stacks were acquired using a Leica SP8 confocal with a 40X immersion objective. Each anterior lymph gland lobe was analysed manually. When more than two cells (Mys$^+$, α-PS4$^-$) or (Mys$^+$, αPS4$^+$) were quantified per anterior lobe, the latter was considered as containing immature (Mys$^+$, αPS4$^-$) or mature (Mys$^+$,αPS4$^+$) lamellocytes, respectively. The percentage of lymph glands corresponds to the number of anterior lobes having lamellocytes divided by the total number of anterior lobes analysed. At least 16 anterior lymph gland lobes were analysed per genotype.

## Statistical analysis

All statistical analyses were performed using R. Encapsulation proportions during artificial selection were arcsine transformed and analysed with a general linear mixed effects model where selection treatment, generation and the interaction term were considered fixed factors and population a random variable. Oil melanization proportions were arcsine transformed and selection regime was used as an explanatory variable in a linear model. To analyse *atilla* and *PPO3* expression from qPCR results we used mixed effects models with treatment, selection regime and the interaction term as fixed effects and population and generation as random variables. Total hemocyte counts and the arcsine transformed lamellocyte proportions were analysed with mixed effects models with treatment and selection regime as explanatory variables and population as a random variable. Crystal cell counts were analysed with a linear mixed effects model with selection regime and sex as explanatory variables and population and generation as random variables. The arcsine square root transformed proportions of cells considered atilla$^+$ with immunohistochemistry staining were analysed with a linear model with treatment, selection regime, and the interaction term as explanatory variables. The disruption state and presence of immature and mature lamellocytes in lymph gland lobes were analysed with mixed effects models with treatment and selection regime as explanatory variables and population and larva as random variables. In all cases, significance was assessed with ANOVA.

## Hemocyte preparation for scRNAseq

Larvae from both selection regimes and infection treatments were obtained following the protocol for hemocyte counting, as described above. 80 to 140 larvae per biological replica were randomly selected, cleaned in PBS and dried on filter paper. Larvae were transferred into 200 µl PBS and dissected by tearing the ventral cuticle. 200 µl of hemocyte solution were collected and filtered (Flowmi cell strainer 70 µm). The cell suspension was carefully transferred into a 15 ml centrifuge tube with OptiPrep solution (Sigma-Aldrich D1556; 540 µl OptiPrep and 1460 µl PBS). Samples were centrifuged at 170 g for 20 min in a swinging bucket centrifuge at 4°C with no brake. The first two layers of 100 µl were then carefully removed and hemocytes were counted in a Thoma chamber. The sample fraction with more hemocytes was used for encapsulation in 10X single-cell platform at Cancer Research UK Cambridge Institute. Library preparation was performed with standard protocols of single-cell 3' V2 chemistry 10X Genomics.

## Read alignments and cell detection

Demultiplexed FASTQ data files were aligned to a custom-built *D. melanogaster* reference obtained from FlyBase FB2018_02 Dmel Release 6.21 (*Thurmond et al., 2019*) and created using the Cell Ranger pipeline (RRID:SCR_017344) (https://support.10xgenomics.com/single-cell-gene-expression/software/overview/welcome, last accessed 23-03-2020). Less than 1,000 cells were detected in libraries generated from control and selected populations in one of the three replicates (replicate two). This replicate was dropped in further analysis. 25,128 cells were identified across the two remaining replicates. The entire sequence of data filtering and clustering is listed in *Figure 3—figure supplement 6*. Most of the analyses were done in R versions 3.6 (*R Development Core Team, 2018*) and

we used the Tidyverse (RRID:SCR_019186), gridExtra and cowplot (RRID:SCR_018081) packages to manipulate and plot the data.

## Normalization, integration, and dimensionality reduction

Seurat version 3.1.1 (*Stuart et al., 2019*) (RRID:SCR_007322) was used to process and analyse the single-cell count matrix. Unique molecular identifier (UMI) count matrices obtained from Cell Ranger were normalized using a scaling factor of 10,000 and log-normal transformation. 2000 highly variable genes (HVG) were discovered using the 'vst' method (*Supplementary file 7*). Count matrices from individual samples were integrated allowing for a maximum of 50 dimensions and 2000 anchor features.

We identified 281 cells that expressed fewer than 250 features or more than 2500 features and 5,387 cells that did not express either *He* or *Srp*, two pan-hemocyte markers (*Banerjee et al., 2019*). These cells were removed from the constituent samples. Following this, the count matrices were normalized, HVG were identified, and the samples were re-integrated.

## Mean fold change in gene expression

To combine data across cells to examine gene expression differences between treatments, we performed the data integration steps utilizing all possible anchors features in Seurat. In doing so, a total of 7716 genes were used for integrating the sample count matrices (*Supplementary file 7*). Fold changes in expression between treatments were obtained from the output of *FindMarkers* with min. pct and logfc.threshold both set to 0. The fold change was converted to a $\log_2$ scale and plotted using the R package LSD. We conduced gene ontology (*Huang et al., 2009*) enrichment analyses on the 173 genes that had at least a 50% change in transcript levels after infection or selection using the Flymine database and tool (*Lyne et al., 2007*) (RRID:SCR_002694) with the 7716 genes used as the background list.

## Clustering cells from scRNA-seq

Cells were clustered in Seurat. First, expression levels were scaled and percent mitochondria, number of features and UMI count per cell were regressed out. Then, we performed PCA and UMAP dimensionality reductions allowing for 50 dimensions. We did a first round of clustering using the 'pca' reduction and a clustering 'resolution' of 0.2. We identified markers upregulated in each cluster using the function *FindAllMarkers* with 'min.pct' and 'logfc.threshold' both set to 0.25 and tested for differential expression using MAST (*Finak et al., 2015*). We identified significantly enriched gene ontology categories (*Huang et al., 2009*) and KEGG (RRID:SCR_012773) and Reactome (RRID:SCR_003485) pathways (*Fabregat et al., 2016*; *Kanehisa et al., 2006*) in each cluster using upregulated cluster markers as query and the 2,000 HVG as the background list.

Enrichment analyses from the first round of clustering indicated one cluster, comprising of 28 cells, was highly enriched for male germ cell expressed genes (*Supplementary file 8* - cluster 9). These cells were removed from the constituent sample count matrices and the data were re-integrated. In the second round of clustering, a 'resolution' of 0.3 was used and we discovered that cluster 9 (48 cells) and cluster 10 (41 cells) were enriched for genes associated with muscle and fat body tissues, respectively (*Supplementary file 9*). These cells were removed from the constituent sample count matrices and data were re-integrated. For round three of clustering, a 'resolution' of 0.3 was utilized. At this point, the 19,344 remaining cells were partitioned into nine clusters.

We identified the *D. melanogaster* orthologs of human cell cycle markers identified in *Tirosh et al., 2016* using the EBI database (*Madeira et al., 2019*) (RRID:SCR_004727). The expression levels of these cell cycle genes were used to classify cells by phase using the function *CellCycleScoring* in Seurat. Regressing out the difference between the G2M and S phases had a minimal impact on the cell cluster classification for all but one cluster (*Supplementary file 10*). The greater discordance among the lamellocyte cluster CC-5 is due to the placement of cells differentiating along a single lineage into discrete clusters.

We tested if using all possible anchors rather than just the highly variable ones had an impact on data integration and clustering. Clustering the data set the 7716 anchors using the same resolution (0.3) largely revealed the same cell clusters save for the appearance of two very small clusters comprising of 113 and 56 cells (cluster 9 and 10 - *Supplementary file 11*). Also, this data set showed

that the proportion of lamellocytes (represented by clusters 1,2,4, and 5) increased following high parasitism and infection (*Figure 3—figure supplement 7*) suggesting the choice of anchors did not result in erroneous patterns. When necessary, we converted the Flybase IDs to gene symbols using the org.Dm.eg.db database v.3.7 (*Carlson, 2018*). The R package reshape2 (*Wickham, 2007*) was used to estimate proportions of cells in each cluster.

## Subclustering lamellocytes

We performed subclustering on lamellocytes and their plasmatocyte progenitors to better resolve the trajectory of cell differentiation. Clusters 1, 2, 4, and 5 from round three clustering had enriched expression of described lamellocyte markers (*ItgaPS4*, *atilla*, *mys*, *PPO3* - *Banerjee et al., 2019*) and, therefore, these cells were isolated for subclustering (*Figure 3—figure supplement 8A*). Cluster 0 from round three clustering was identified as the progenitor plasmatocyte population based on the expression of five plasmatocyte markers (*Col4a1*, *Pxn*, *eater*, *Hml*, *NimC1* - *Banerjee et al., 2019*) and three cell cycle markers (*stg* – *Edgar and O'Farrell, 1990*, *polo* – *Glover, 2005*) and *scra* – (*Oegema et al., 2000*; *Figure 3—figure supplement 8B*).

The presumed lamellocytes and plasmatocyte progenitor were subset from the constituent sample count matrices. The cell count matrices for each sample were normalized and 2,000 HVG were found. Following this, dimensionality reduction and round one of subclustering was performed using a 'resolution' parameter of 0.3. In round one of subclustering, we identified 79 cells that were significantly enriched for markers representing crystal cells (*Figure 3—figure supplement 8C* and *Supplementary file 12* - cluster 5). These cells were moved to the crystal cell cluster, and we did a second round of subclustering on the remaining cells. We followed the same data integration and clustering steps but used a resolution of 0.2. The original cluster identities for lamellocytes and plasmatocyte progenitor cells were replaced with their new identities inferred from the subclustering.

## Trajectory inference

We used the software Slingshot (*Street et al., 2018*) (RRID:SCR_017012) to conduct pseudotime analysis. We used all 50 PCA dimensions estimated from Seurat for trajectory inference and ensured lineages were not inferred past endpoints by setting stretch to 0. The cluster with the highest expression of the three cell cycle markers (*stg*, *polo* and *scra*) was set as starting cluster.

We identified genes that were differentially expressed between the starting and terminal clusters of lamellocyte differentiation (PLASM1 vs LAM3). We performed gene ontology enrichment on genes with log(e)FC >1 using the *goana* function, part of the R package 'limma' (*Ritchie et al., 2015*) (RRID:SCR_010943) with the 2000 HVG used as the background list. Gene set enrichment analyses were conducted separately on markers upregulated in the starting (PLASM1) and terminal clusters (LAM3). We removed redundant gene ontology terms using REVIGO (*Supek et al., 2011*) (RRID:SCR_005825) limiting false discovery rate adjusted P-values to 0.05 and similarity to 0.4. We produced heatmaps of the variable genes belonging to significantly enriched gene ontology categories using a custom modified version of the *DoHeatmap* function from Seurat v3 and grouped genes by mean cluster expression levels. For the heatmap, the log2 fold change gene expression levels were centred on their mean value and divided by their standard deviation. Smaller gene ontology categories whose membership consisted of genes that were wholly part of larger categories were not included in the heatmap plots.

The 2000 HVG were also utilized for identifying lineage markers. A random forest model was fit to identify the genes best able to predict the pseudotime values using R tidymodels. The R package 'Ranger' (*Wright and Ziegler, 2017*) was used for implementation of the random forest using 2000 trees and a min_n of 15. Variable importance was evaluated by the 'altmann' test (*Altmann et al., 2010*) using 100 permutations. We produced a heatmap of the top 100 lineage markers ranked by variable importance, as described in the preceding section.

## Cluster definitions

We performed pathway enrichment analyses on all non-lamellocyte clusters with the aim of defining them according to their inferred biological function. First, upregulated cluster markers were identified through pairwise differential expression tests. The cluster of interest was compared with PLASM1 using the function *FindMarkers* with 'min.pct' and 'logfc.threshold' both set to 0.25. Genes

with significant upregulation in each cluster were identified using the MAST test statistic (*Supplementary file 13*) after performing Bonferroni correction on the obtained P-values. Markers upregulated in each cluster were used for pathway enrichment analysis. Markers for the plasmatocyte progenitor population were identified through comparisons to the cell types containing other plasmatocyte-type cells. These markers were used for pathway enrichment analyses using the Flymine database and tool (*Lyne et al., 2007*).

## Comparison to other scRNA-seq datasets

We compared our clusters of cells with clusters defined in two published scRNA-seq data sets generated from *D. melanogaster* larval hemocytes. We attained the expression and cell cluster data for *Tattikota et al., 2020* from the gene expression omnibus (GEO) repository (RRID:SCR_005012) under accession GSE146596. We attained the expression and cell cluster data for *Cattenoz et al., 2020* from the ArrayExpress repository (RRID:SCR_002964) under accession E-MTAB-8698. We used the expression and cluster identity data to create Seurat v3 objects and used Flybase IDs as feature names. For *Cattenoz et al., 2020*, we combined the the WI and NI data sets into a single object. We normalized the counts, identified 2000 HVG for each data set and scaled the data using the default options in Seurat v3. We performed a reference-based label transfer using the *FindTransferAnchors* and *TransferData* functions with a 'cca' reduction and L2 normalization. We compared the three data sets in pairs and alternated the reference and query for each comparison. We plot the proportion of cells in query clusters that were predicted to belong to a given reference cluster using the R package 'pheatmap' (RRID:SCR_002630).

## Acknowledgements

The Genomics Core Facility at Cancer Research UK Cambridge Institute, G Raddi and S Tattikota provided protocols for the scRNAseq.

## Additional information

### Funding

| Funder | Grant reference number | Author |
| --- | --- | --- |
| Natural Environment Research Council | NE/P00184X/1 | Alexandre B Leitão<br>Francis M Jiggins |
| EMBO | ALT-1556 | Alexandre B Leitão |
| Natural Sciences and Engineering Research Council of Canada | PDF-516634-2018 | Ramesh Arunkumar |

The funders had no role in study design, data collection and interpretation, or the decision to submit the work for publication.

### Author contributions

Alexandre B Leitão, Conceptualization, Formal analysis, Funding acquisition, Investigation, Writing - original draft; Ramesh Arunkumar, Conceptualization, Data curation, Formal analysis, Writing - original draft; Jonathan P Day, Investigation, Writing - review and editing; Emma M Geldman, Ismaël Morin-Poulard, Investigation; Michèle Crozatier, Conceptualization, Supervision, Writing - review and editing; Francis M Jiggins, Conceptualization, Supervision, Funding acquisition, Writing - original draft, Project administration

### Author ORCIDs

Ramesh Arunkumar https://orcid.org/0000-0003-0050-3516
Michèle Crozatier http://orcid.org/0000-0001-9911-462X
Francis M Jiggins https://orcid.org/0000-0001-7470-8157

Decision letter and Author response
Decision letter https://doi.org/10.7554/eLife.59095.sa1
Author response https://doi.org/10.7554/eLife.59095.sa2

## Additional files

### Supplementary files
• Supplementary file 1. Gene ontology enrichment for 173 genes that had >50% change in transcript levels after infection or selection.

• Supplementary file 2. Enriched gene ontology categories and pathways for non-lamellocyte clusters.

• Supplementary file 3. 10x v2 library sequencing metrics.

• Supplementary file 4. Lamellocyte lineage classification with and without cell cycle correction.

• Supplementary file 5. List of marker genes for predicting lamellocyte differentiation.

• Supplementary file 6. Enriched gene ontology categories and pathways for lamellocytes.

• Supplementary file 7. Genes detected in scRNA-seq dataset.

• Supplementary file 8. Cluster markers identified from round one of clustering.

• Supplementary file 9. Cluster markers identified from round two of clustering.

• Supplementary file 10. Hemocyte classification with and without cell cycle correction.

• Supplementary file 11. Hemocyte classification using 2000 highly variable genes or all 7716 detected genes.

• Supplementary file 12. Cluster markers identified from round one of subclustering.

• Supplementary file 13. Marker genes for non-lamellocyte clusters.

• Transparent reporting form

### Data availability
Unprocessed single cell sequence reads were deposited in the Sequence Read Archive (accession: SRP256887, Bioproject: PRJNA625925). Cell count matrices for all detected genes, cluster identities and processed scRNA-seq results were deposited into Gene Expression Omnibus (accession: GSE148826). The R script used to analyze the scRNA-seq data is available on Github (https://github.com/arunkumarramesh/dmel_scRNA_hemocyte; copy archived at https://archive.softwareheritage.org/swh:1:rev:6be24fd50d52e1a95baebb4228979d7223d141ae/).

The following datasets were generated:

| Author(s) | Year | Dataset title | Dataset URL | Database and Identifier |
|---|---|---|---|---|
| Leitão A, Arunkumar R, Day J, Jiggins F | 2020 | Single cell sequencing reveals that constitutive activation of cellular immunity underlies the evolution of resistance to infection | https://www.ncbi.nlm.nih.gov/geo/query/acc.cgi?acc=GSE148826 | NCBI Gene Expression Omnibus, GSE148826 |
| Arunkumar R, Leitão AB, Jiggins FM | 2020 | Constitutive activation of cellular immunity underlies the evolution of resistance to infection in *Drosophila* | https://www.ncbi.nlm.nih.gov/sra/?term=SRP256887 | NCBI Sequence Read Archive, SRP256887 |
| Arunkumar R, Leitão AB, Jiggins FM | 2020 | **Constitutive activation of cellular immunity underlies the evolution of resistance to infection in *Drosophila*** | https://www.ncbi.nlm.nih.gov/bioproject/PRJNA625925 | NCBI BioProject, PRJNA625925 |

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
