## [Decision Letter]

**Acceptance summary:**

Your work demonstrates how activation of the immune system may evolve from inducible to constitutive when organisms are experiencing a high risk of infection for a prolonged number of generations. A nice touch is how this study exploits high-resolution, single cell comparisons of blood cell populations and their gene expression profiles before and after selection for increased immunity.

**Decision letter after peer review:**

Thank you for submitting your article "Constitutive activation of cellular immunity underlies the evolution of resistance to infection in *Drosophila*" for consideration by *eLife*. Your article has been reviewed by three peer reviewers, including Bruno Lemaître as the Reviewing Editor and Reviewer #1, and the evaluation has been overseen by Detlef Weigel as the Senior Editor. The following individual involved in review of your submission has agreed to reveal their identity: Bregje Wertheim (Reviewer #2).

The reviewers have discussed the reviews with one another and the Reviewing Editor has drafted this decision to help you prepare a revised submission.

Summary:

It has been previously shown that resistance to parasitoid wasps can emerge upon selection in wild-type *Drosophila* populations, and that this increased resistance correlates with a higher number of hemocytes. This paper combined experimental evolution and single cell transcriptomics to show that increased resistance to parasitoids upon several rounds of selection is caused by the presence of a differentiated subset of hemocytes (pre-lamellocyte) in the unchallenged state, which is usually found only upon wasp infestation. This led the authors to conclude that intense pathogen pressures can shift the immune system from inducible to constitutive, consistent with a theoretical framework indicating that elevated and constant pathogen pressure should lead to the emergence of constitutive defense. The approach is interesting, the paper well-written and the notion tested interesting. An important concern is the degree of advance over previous studies. Initial papers investigating how selection increases resistance to wasps have already shown that this was linked to an increase in hemocyte number. In a certain sense, this could be considered as a demonstration of a change from inducible to constitutive defense, although the emphasis of these papers was not on this point. In addition, the current work provides so far only limited information on this specific population of pre-lamellocytes.

Essential revisions:

1) Analysis of single cell RNA-seq (Figure 3). Several RNA-seq papers have been published and it is important that the authors better relate their hemocyte clusters to other scRNA-seq datasets using the nomenclature of some of these papers. Would their data be deposited in a database? It would also be great to better describe the transcriptional profile, and not only focus on two genes, Attila and *PPO3*.

2) Discrepancy between the transcriptional and morphological changes in the hemocytes

a) Earlier studies, both on hemocyte flow cytometry and in other scRNA-seq experiments (as cited in the manuscript) revealed that the transdifferentiation into lamellocytes is a dynamic / continuous process, which may derive from several hemocyte lineages and from different hematopoietic organs. The authors here showed a discrepancy in the transcriptional and morphological changes in the hemocytes, and revealed that the plasmatocyte lineage was already starting the resemble the lamellocytes (in gene expression), without needing the induction by infection. Yet, they were not yet fully differentiated hemocytes based on morphology, and still needed infection to reach that stage. Therefore, the conclusion that the selected lines had "hard-wired" the inducible response into a constitutive response is not fully warranted (they do not fully differentiate, but proceed partially towards that state). Also, the differentiation of lamellocytes is fully attributed to originate from lymph glands and as originating from the plasmatocytes, while different organs and hemocyte lineages appear to contribute to the population of lamellocytes. The reviewer feel that all these aspects should be further explored and would deserve some mentioning in the Discussion.

b) Along this line, the authors could do a better job in characterizing the hemocyte populations of the evolved lines using available antibody, cooking and other melanization assays, phalloidin treatment…

c) Third instar hemocytes are found in the sessile state, in circulation or in the lymph gland. It could not be excluded that some of the changes they observed relate more to changes in hemocyte localization rather than differentiation. According to the Materials and methods, the authors has collected only the circulating hemocytes in the unchallenged state as they did not vortex larvae. It is very important to better compare the lymph gland, sessile and circulating compartments of the evolved and the non-evolved lines. This can be done by using various staining methods. The paper is written in such a way that selection acted only on circulating hemocytes but it could also act on hemocytes localization (decrease sessility), lymph gland maturation.

3) Gene expression was measured in circulating hemocytes at 48h after infection

The authors measured gene expression in circulating hemocytes, 48h after infection, at which stage hemocyte proliferation, lamellocyte differentiation and parasitoid encapsulation is already well underway. The induction of the critical two processes, hemocyte proliferation and lamellocyte differentiation, may not be fully detectable from gene expression of only the circulating hemocytes themselves at this late stage of the immune response. Clearly, the authors do show that differentiation from circulating plasmatocytes can be detected, using pseudotime, and also revealed changes in gene expression in uninfected selected larvae. Yet, how induction in the lymph glands or sessile clusters has changed by experimental evolution, and whether the inducible response had indeed proceeded towards a constitutive response, requires further investigation along a wider time course (e.g. during early larval development) and perhaps in different tissues (e.g. lymph glands). If the author cannot address this, this aspect would need some discussion.

4) The changes in gene expression after selection can be presented clearer

The description of these results (subsection “Constitutive upregulation of immune-induced genes”), and Figure 2, are difficult to read and understand, while it is key to the claim that the inducible response has become hardwired into a constitutive response. In the text it starts out with saying that "data was pooled to investigate global changes", but then it refers in Figure 2 to the x-axis which only provides the data for the control lines. This figure 2 is difficult to grasp, as the strong positive correlation in a) means something different (i.e. stronger constitutive response) than the very similar positive correlation in b) (weaker induced response), while c shows that control and selected larvae respond the same to infection. Is there a better way to tease apart these patterns in a figure, and to explain them in the text? Also, the data is all expressed in log2 fold changes (relative to non-infected control line individuals?). Also, for a subset of approximately 170 genes, the authors showed that the increase in expression had already started without the infection in the selection lines. Do the functional annotations of these genes reveal anything of interest for hemocyte proliferation and the differentiation towards lamellocytes?

5) Other studies came to partially contrasting, partially similar conclusions.

Transcriptomics on whole larvae after experimental evolution for high parasitism was done for *Drosophila*, using a different parasitoid species. In this study, they also found the typical increased density of hemocytes in *Drosophila* selected for increased parasitoid resistance, without being infected. However, contrary to the authors, this study concluded this increase in hemocytes could not be attribute to a pre-activation of the immune response. Additionally, the genes for hematopoiesis and for several effector genes showed opposite patterns to those that would explain the increased density of hemocytes in selected lines or for an pre-activation of the inducible response (Wertheim et al., 2011). However, in line with the findings for the current study, whole-larvae RNAseq after parasitoid infection did not result in substantial gene expression differences between selected lines and control lines (Salazar-Jaramillo et al., 2017), while substantial differences were reported in uninfected larvae of selection and control line larvae (Wertheim et al., 2011). These whole-body transcriptomics experiments lacked the resolution to measure specifically what changed in hemocytes, but both studies indicate that much of the increased resistance after selection is likely caused by changes in constitutive immunity, not by increasing the acute/inducible immune response.

6) Another concern is related to the parasitoid species. *Leptopilina boulardi* is a parasitoid that relies partly on VLPs to overcome the host defense. This is not discussed, not even mentioned. Some older work (Fellowes et al. 1999, Evolution), shows that, while resistance evolves readily against *L. boulardi*, populations resistant against *L. boulardi* are also cross-resistant to another *Leptopilina* species. The immune effectors studied in this manuscript are obviously playing a significant role, but how do the evolved flies cope with the VLPs? The paper would benefit from at least discussing this issue.

7) The selection of larvae for the single cell work warrants some clarification. According to Figure 1B just under 50% of parasitoid resistant larvae showed an increased encapsulation response. This is presumably also related to the increase of expression of immune effectors. How is this accounted for in the single cell work? And if not, do you have any way to get an estimate of the variance in the response variables?

---

## [Author Response]

Essential revisions:1) Analysis of single cell RNA-seq (Figure 3). Several RNA-seq papers have been published and it is important that the authors better relate their hemocyte clusters to other scRNA-seq datasets using the nomenclature of some of these papers.

We have now performed a new analysis to compare our scRNA-seq data to the two published studies that made their data publicly available (Cattenoz et al., 2020; Tattikota et al., 2020). In addition to lamellocytes, the crystal cell cluster (CC) and antimicrobial peptide (AMP) cluster had a high degree of similarity among all three data sets. Hence, we have used the same acronyms to define CC and AMP clusters in our paper. This is not the case for plasmatocytes, so we have left the names unaltered. All three studies identify similar groups of mature and immature lamellocytes. However, this is a continuum of cells and the studies split them into different numbers of clusters, so again we have stuck with our own names.

We have added the following text:

Results: “Two similar scRNA-seq studies of *D. melanogaster* larval hemocytes have recently been published, so we investigated whether they identified similar clusters of cells to us (Cattenoz et al., 2020; Tattikota et al., 2020). […] However, there was less correspondence between studies for the other clusters of plasmatocytes (Figure 3—figure supplement 3).”

Results: “Together, these results suggest that LAM1 and LAM2 cells are immature lamellocytes, while LAM3 cells are mature lamellocytes. […] Similarly, the immature lamellocyte clusters, LAM1 and 2, match the immature lamellocyte clusters in those studies (Cattenoz et al., 2020 – LM-2; Tattikota et al., 2020 – LM1) (Figure 3—figure supplement 3).”

Materials and methods: “Comparison to other scRNA-seq datasets

We compared our clusters of cells with clusters defined in two published scRNA-seq data sets generated from *D. melanogaster* larval hemocytes. […] We plot the proportion of cells in query clusters that were predicted to belong to a given reference cluster using the R package “pheatmap”.”

Would their data be deposited in a database?

Yes, our scRNA-seq data has been submitted to the gene expression omnibus (GEO) repository under accession: GSE148826 and will be made publicly available upon acceptance of our manuscript. We have provided a private access key for editors and reviewers to access the data in our cover letter.

The results read:

“Unprocessed single cell sequence reads were deposited in the Sequence Read Archive (accession: SRP256887, Bioproject: PRJNA625925). […] The R script used to analyze the scRNA-seq data is available on Github (Repository: dmel_scRNA_hemocyte).”

It would also be great to better describe the transcriptional profile, and not only focus on two genes, Attila and PPO3.

We have now performed a gene ontology enrichment analysis for genes that are upregulated in the starting (PLASM1) and terminal (LAM3) clusters of lamellocyte differentiation. We have also included a heatmap of the top 100 markers predicting lamellocyte differentiation.

We have also performed a gene ontology enrichment analysis for the 173 genes that have a 50% change in transcript levels after infection or selection. The majority of these genes were also found to significantly upregulated in LAM3 and are enriched in similar biological function categories (Supplementary file 1, text below).

Our descriptions are *atilla* and *PPO3* effecter genes were meant highlight the suitability of the markers to track lamellocyte differentiation in the scRNA-seq data. Their association with the differentiation has been previously well-established (Banerjee et al., 2019). We used these two genes to correlate how differentiation as inferred from scRNA-seq correlates with changes in lamellocyte morphology (Figure 5).

Results (bulk RNAseq): “[Taking the 173 genes that had at least a 50% change in transcript levels after infection or selection, in every case the direction of change was the same after infection and adaptation to high parasitism (Figure 2B).] These genes were enriched for biological roles such as cytoskeletal organization and cell adhesion (Supplementary file 1).”

Results: “The expression of numerous genes changed as cells progressed along this trajectory (Figure 4—figure supplement 1C, Supplementary file 5), including increases in the expression of the lamellocyte effector gene *PPO3* and the lamellocyte marker *atilla* (Figure 4B, C). […] This likely reflects the loss of the ability of cells to secrete extracellular matrix, which is a house-keeping function of plasmatocytes (Olofsson and Page, 2005; Bunt et al., 2010).”

Materials and methods: “We identified genes that were differentially expressed between the starting and terminal clusters of lamellocyte differentiation (PLASM1 versus LAM3). […] Smaller gene ontology categories whose membership consisted of genes that were wholly part of larger categories were not included in the heatmap plots.”

2) Discrepancy between the transcriptional and morphological changes in the hemocytesa) Earlier studies, both on hemocyte flow cytometry and in other scRNA-seq experiments (as cited in the manuscript) revealed that the transdifferentiation into lamellocytes is a dynamic / continuous process, which may derive from several hemocyte lineages and from different hematopoietic organs. The authors here showed a discrepancy in the transcriptional and morphological changes in the hemocytes, and revealed that the plasmatocyte lineage was already starting the resemble the lamellocytes (in gene expression), without needing the induction by infection. Yet, they were not yet fully differentiated hemocytes based on morphology, and still needed infection to reach that stage. Therefore, the conclusion that the selected lines had "hard-wired" the inducible response into a constitutive response is not fully warranted (they do not fully differentiate, but proceed partially towards that state). Also, the differentiation of lamellocytes is fully attributed to originate from lymph glands and as originating from the plasmatocytes, while different organs and hemocyte lineages appear to contribute to the population of lamellocytes. The reviewer feel that all these aspects should be further explored and would deserve some mentioning in the Discussion.

In the Abstract we now clarify the term “hard-wired”: “Therefore, populations evolved resistance by genetically hard-wiring the first steps of an induced immune response to become constitutive.”

Results: “Therefore, as populations evolved resistance, the induced transcriptional response to infection has become in part genetically hard-wired and constitutive.”

We address the comment about lymph glands under point 2c below.

b) Along this line, the authors could do a better job in characterizing the hemocyte populations of the evolved lines using available antibody, cooking and other melanization assays, phalloidin treatment…

We now include a count of sessile crystal cells. The results are in agreement with the other observations made now in response to point 2c that there was no relocation of sessile hemocytes into circulation after selection with high parasitism. We also tried crossing our lines to lines expressing hemocyte reporters and sorting cells by flow cytometry, but this was not successful as the F1 progeny had greatly reduced resistance:

Results: “In accordance with this result, the number of crystal cells in the sessile compartment was not altered in populations maintained with high parasitism (Figure 5—figure supplement 6; Selection regime: χ^2^=0.014, d.f=1, *p*=0.91).”

c) Third instar hemocytes are found in the sessile state, in circulation or in the lymph gland. It could not be excluded that some of the changes they observed relate more to changes in hemocyte localization rather than differentiation. According to the Materials and methods, the authors has collected only the circulating hemocytes in the unchallenged state as they did not vortex larvae. It is very important to better compare the lymph gland, sessile and circulating compartments of the evolved and the non-evolved lines. This can be done by using various staining methods. The paper is written in such a way that selection acted only on circulating hemocytes but it could also act on hemocytes localization (decrease sessility), lymph gland maturation….

We focused our study in circulating hemocytes because it is the compartment where they are essential to form the capsule around parasitoid eggs.

Certainly, the physiological changes that lead to a change in this compartment are important to understand. We have now included two major new experiments to address the questions raised by the reviewers.

To test if artificial selection with high parasitism increased the total number of hemocytes or their localization, we have measured hemocyte concentration after vortexing larvae with glass beads, a treatment that moves sessile hemocytes into circulation (Petraki, Alexander and Brückner, 2015). The results show that selection with high parasitism led to a higher production of hemocytes in the circulating and sessile compartments.

Results: “We found that populations maintained under the high parasitism regime increased the total number of hemocytes, rather than move cells from the sessile compartment into circulation (Figure 5—figure supplement 5; Selection regime: χ^2^=34.79, d.f=1, *p*=3.67x10^-^ 9; Selection regime * Compartment: χ^2^=2.41, d.f=1, *p*=0.12).”

Materials and methods: “To relocate hemocytes from sessile compartments into circulation, groups of 10 larvae were transferred into a 2ml microcentrifuge tube with 0.5ml water and 0.5g of glass beads (425- 600μm, Sigma #G9268). The tubes were then vortexed at maximum speed for 1min (Petraki et al., 2015) and larvae prepared for hemolymph collection as described above.”

To test if artificial selection with high parasitism changed lymph gland maturation we collaborated with Michèle Crozatier and Ismael Morin-Poulard. Antibodies anti *alpha-PS4* and *mys* were used to detect immature (α-PS4^-^ mys^+^) and mature lamellocytes (α-PS4^+^ mys^+^). The results show that, similar to what happens in circulating hemocytes, lymph glands from larvae selected with high parasitism produce more immature lamellocytes in homeostasis.

Results: “Lamellocytes can differentiate from sessile or circulating plasmatocytes, or from cells in the lymph gland, which is the larval hematopoietic organ. […] Therefore, adaptation to high parasitism leads to an increase in both the differentiation of lamellocyte precursors in the lymph gland and the premature disruption of lymph glands.”

Materials and methods: “Lymph glands were dissected and processed as previously described (Krzemień et al., 2007). […] At least 16 anterior lymph gland lobes were analyzed per genotype.”

3) Gene expression was measured in circulating hemocytes at 48h after infectionThe authors measured gene expression in circulating hemocytes, 48h after infection, at which stage hemocyte proliferation, lamellocyte differentiation and parasitoid encapsulation is already well underway. The induction of the critical two processes, hemocyte proliferation and lamellocyte differentiation, may not be fully detectable from gene expression of only the circulating hemocytes themselves at this late stage of the immune response. Clearly, the authors do show that differentiation from circulating plasmatocytes can be detected, using pseudotime, and also revealed changes in gene expression in uninfected selected larvae. Yet, how induction in the lymph glands or sessile clusters has changed by experimental evolution, and whether the inducible response had indeed proceeded towards a constitutive response, requires further investigation along a wider time course (e.g. during early larval development) and perhaps in different tissues (e.g. lymph glands). If the author cannot address this, this aspect would need some discussion.

The effect of artificial selection in the induction of lamellocytes in the lymph gland is addressed above, in point 2c. This experiment was challenging as we are only allowed to work in the morning due to covid. To understand the dynamic differentiation of lamellocytes, we have now included a description of hemocyte composition during development in homeostasis and during infection. The results show that larvae from populations selected with high parasitism produce lamellocytes more rapidly after infection.

Materials and methods: “To count hemocytes during the course of experimental evolution, vials were incubated 48h post infection. To estimate hemocyte numbers during the course of infection, vials were incubated for 12, 24, 36 and 48h post infection.”

Results: “Based on our analyses of both circulating hemocytes and lymph glands, we hypothesised that accelerated lamellocyte production during infection underlies the evolution of resistance. […] Altogether, these results support the hypothesis that constitutive activation of immune defences allows hosts to mount more rapid immune responses.”

4) The changes in gene expression after selection can be presented clearerThe description of these results (subsection “Constitutive upregulation of immune-induced genes”), and Figure 2, are difficult to read and understand, while it is key to the claim that the inducible response has become hardwired into a constitutive response. In the text it starts out with saying that "data was pooled to investigate global changes", but then it refers in Figure 2 to the x-axis which only provides the data for the control lines. This Figure 2 is difficult to grasp, as the strong positive correlation in a) means something different (i.e. stronger constitutive response) than the very similar positive correlation in b) (weaker induced response), while c shows that control and selected larvae respond the same to infection. Is there a better way to tease apart these patterns in a figure, and to explain them in the text?

We thank the reviewer for pointing out the lack of clarity of Figure 2 and have made two changes to make this key figure easier to understand. First, we have removed what was panel B in the previous submission, where we compared the induced response in high and no parasitism populations. On reflection, we believe this can be inferred from the remaining panels. Second, we have included a schematic in panel A depicting and numbering the comparisons we have performed. We hope this reiteration is clearer in showing that the pattern of gene expression in populations selected under high parasitism is similar to the pattern of gene expression observed following wasp infection in populations that did not undergo parasitoid selection pressures.

Also, the data is all expressed in log2 fold changes (relative to non-infected control line individuals?).

Yes. This is indicated in each axis label in Figure 2.

Also, for a subset of approximately 170 genes, the authors showed that the increase in expression had already started without the infection in the selection lines. Do the functional annotations of these genes reveal anything of interest for hemocyte proliferation and the differentiation towards lamellocytes?

A gene ontology enrichment analysis indicated that these 173 genes (Supplementary file 1) were enriched for many of the same biological function categories as genes that were upregulated in mature lamellocytes (LAM3 – Supplementary file 6). This is because many of the 173 genes were upregulated as lamellocytes differentiated and matured. We appreciate this comment as it shows that the “hardwiring” of transcription response to infection in selected populations is due to the constant differentiation of (pre)lamellocytes even in the absence of infection.

The revisions we made in response to this are detailed as part of our response to comment 1).

5) Other studies came to partially contrasting, partially similar conclusions.Transcriptomics on whole larvae after experimental evolution for high parasitism was done for Drosophila, using a different parasitoid species. In this study, they also found the typical increased density of hemocytes in Drosophila selected for increased parasitoid resistance, without being infected. However, contrary to the authors, this study concluded this increase in hemocytes could not be attribute to a pre-activation of the immune response. Additionally, the genes for hematopoiesis and for several effector genes showed opposite patterns to those that would explain the increased density of hemocytes in selected lines or for an pre-activation of the inducible response (Wertheim et al., 2011). However, in line with the findings for the current study, whole-larvae RNAseq after parasitoid infection did not result in substantial gene expression differences between selected lines and control lines (Salazar-Jaramillo et al., 2017), while substantial differences were reported in uninfected larvae of selection and control line larvae (Wertheim et al., 2011). These whole-body transcriptomics experiments lacked the resolution to measure specifically what changed in hemocytes, but both studies indicate that much of the increased resistance after selection is likely caused by changes in constitutive immunity, not by increasing the acute/inducible immune response.

We thank the authors to point to this omission in our Discussion. The differences between studies are important to discuss, although the causes for these differences can be very diverse and worth to explore in future experiments. Firstly, the wasp species used can have different impacts on composition of hemocytes since the genetic basis of resistance is, partially, different for the two species. This is evident because lines resistant for *L. boulardi* are cross-resistant to *A. tabida* but the opposite is not true (Fellowes, Kraaijeveld and Godfray, 1999). It is possible that this phenotype is specifically selected in lines selected for *L. boulardi.* Another difference is the type of sample analysed. Selection has an effect on hemocyte composition but this is still a subtle change, in comparison with the changes caused by infection. When the whole larva body is used to analyse gene expression the power to detect this differences may be too reduced. To add to this caveat, lines have a higher number of total hemocytes, a phenotype that can be detected with whole body RNA extraction but cannot be studied in our analysis, which can just compare differences between hemocyte populations.

To address these points we added the following text:

Discussion: “Our results contrast with populations evolved with high levels of parasitism by the parasitoid wasp *Asobara tabida*, where there was no significant overlap of differentially expressed genes between selection and infection (Wertheim et al., 2011). […] The could also result from the different genetic basis of resistance to the two wasp species (Poirie et al., 2000).”

“Furthermore, the late stages of the immune response was not qualitatively altered after selection. Populations from both selection regimes had very similar hemocyte compositions in the late stages of the response, which is similar to patterns reported from populations artificially selected with *Asobara tabida* (Salazar-Jaramillo et al., 2017).”

6) Another concern is related to the parasitoid species. Leptopilina boulardi is a parasitoid that relies partly on VLPs to overcome the host defense. This is not discussed, not even mentioned. Some older work (Fellowes et al. 1999, Evolution), shows that, while resistance evolves readily against L. boulardi, populations resistant against L. boulardi are also cross-resistant to another Leptopilina species. The immune effectors studied in this manuscript are obviously playing a significant role, but how do the evolved flies cope with the VLPs? The paper would benefit from at least discussing this issue.

We now include in the Discussion section the following discussion about VLPs:

“To increase their success, parasitoid wasps inject venom proteins and virus like particles (VLPs) during oviposition to suppress the host immune system (Labrosse et al., 2003). […] Thus, resistant populations must circumvent the suppressive effects of the wasp venoms and VLPs, and constitutive activation of cellular immunity may be a way to achieve this by killing the parasite before immunity is suppressed.”

7) The selection of larvae for the single cell work warrants some clarification. According to Figure 1B just under 50% of parasitoid resistant larvae showed an increased encapsulation response. This is presumably also related to the increase of expression of immune effectors. How is this accounted for in the single cell work? And if not, do you have any way to get an estimate of the variance in the response variables?

The larvae from infection treatments were selected randomly, i.e., it included larvae with capsules and without capsules. This is now made clear in the Materials and methods section “80 to 140 larvae per biological replica were randomly selected”. This was a limitation of our protocol that required a very large number of larvae being dissected in a short period of time to have enough hemocytes.

Unfortunately, we have neither have data to explore the genetic basis of resistance in these lines, nor any way to know is cells came from larvae that had an encapsulation response. Without it, it becomes speculative to address this point. For example, parasitoid infections in isogenic lines can result in intermediate proportion of larvae with capsules (Leitão et al., 2019), so there may not be genetic differences between larvae with and without an encapsulation response.